# Tissue-specific responses to TFAM and mtDNA copy number manipulation in prematurely ageing mice

Laura Sophie Kremer[1,2]*[†], Guanbin Gao[1][†], Giovanni Rigoni[1], Roberta Filograna[1], Mara Mennuni[1], Rolf Wibom[3], Ákos Végvári[1], Camilla Koolmeister[1], Nils-Göran Larsson[1]*

[1]Department of Medical Biochemistry and Biophysics, Karolinska Institutet, Stockholm, Sweden; [2]Department of Cellular Biochemistry, University Medical Center Göttingen, Göttingen, Germany; [3]Center for Inherited Metabolic Diseases, Karolinska University Hospital, Stockholm, Sweden

## eLife Assessment

This is an **important** study that examines the role of TFAM, a protein that helps maintain mtDNA, in mtDNA mutator mice. With **convincing** evidence, the authors have demonstrated that TFAM's counteractive role in mtDNA mutator mice is tissue-specific. The study does a thorough job of assessing the impact of modulating TFAM levels in a polg mutator mouse model of aging. The authors have thoroughly addressed all the points raised during the first round of review.

*For correspondence:
laura.kremer@med.uni-goettingen.de (LSK);
nils-goran.larsson@ki.se (N-GL)

[†]These authors contributed equally to this work

**Abstract** Somatic mitochondrial DNA (mtDNA) mutations are implicated as important drivers of ageing and age-related diseases. Their pathological effect can be counteracted by increasing the absolute amount of wild-type mtDNA via moderately upregulating TFAM, a protein important for mtDNA packaging and expression. However, strong TFAM overexpression can also have detrimental effects as it results in mtDNA hypercompaction and subsequent impairment of mtDNA gene expression. Here, we have experimentally addressed the propensity of moderate TFAM modulation to improve the premature ageing phenotypes of mtDNA mutator mice, carrying random mtDNA mutations. Surprisingly, we detect tissue-specific endogenous compensatory mechanisms acting in mtDNA mutator mice, which largely affect the outcome of TFAM modulation. Accordingly, moderate overexpression of TFAM can have negative and beneficial effects in different tissues of mtDNA mutator mice. We see a similar behavior for TFAM reduction, which improves brown adipocyte tissue homeostasis, while other tissues are unaffected. Our findings highlight that the regulation of mtDNA copy number and gene expression is complex and causes tissue-specific effects that should be considered when modulating TFAM levels. Additionally, we suggest that TFAM is not the sole determinant of mtDNA copy number in situations where oxidative phosphorylation (OXPHOS) is compromised, but other important players must be involved.

## Introduction

Ageing in animals is widely believed to be caused by the accumulation of multiple types of damage, and from an evolutionary point of view, there has been little selective pressure to maintain efficient repair and maintenance of tissues at a post-reproductive old age (*Kirkwood, 2005*). Although the decline of multiple pathways plays an important role (*López-Otín et al., 2013*), the decrease of mitochondrial function is a prominent feature of ageing (*Bratic and Larsson, 2013*; *Larsson, 2010*). In

particular, somatic mutagenesis of mtDNA is heavily associated with mammalian ageing (**Larsson, 2010**). Notably, low levels of evenly distributed mtDNA mutations may not influence the ageing process (**Vermulst et al., 2007**; **Ameur et al., 2011**), but rather the spatially restricted accumulation of pathogenic mtDNA mutations to high levels by clonal expansion in individual cells (**Sanchez-Contreras and Kennedy, 2021**). In ageing humans, such accumulation of mtDNA mutations results in focal respiratory chain dysfunction in a subset of cells in various tissues, e.g., in the brain, heart, skeletal muscle, and colonic crypts (**Bender et al., 2006**; **Müller-Höcker, 1989**; **Müller-Höcker, 1990**; **Taylor et al., 2003**). While it was long unclear whether mtDNA mutations are a mere consequence of the ageing process or contribute to cell loss and organ dysfunction, the causal role of somatic mtDNA mutations in ageing was established using genetic mouse models. In the mtDNA mutator mouse model, the expression of an exonuclease-deficient catalytic subunit of mitochondrial DNA polymerase (POLG$^{D257A}$; genotype: *Polg*$^{mut/mut}$) leads to extensive accumulation of mtDNA mutations causing a variety of premature ageing symptoms that drastically shorten life span (**Trifunovic et al., 2004**; **Kujoth et al., 2005**). However, it should be noted that the importance of mtDNA as a driver of ageing varies in metazoans and seems not to affect the life span in short-lived organisms. In *Caenorhabditis elegans,* life span is not affected if mtDNA replication is abrogated by knockout of *Polg* (**Bratic et al., 2009**), and in *Drosophila melanogaster,* a massive accumulation of somatic mutations of mtDNA caused by the expression of an exonuclease-deficient version of POLG does not limit life span (**Kauppila et al., 2018**). The somatic mtDNA mutations associated with mammalian ageing are mostly generated during the extensive mtDNA replication that occurs during embryogenesis and postnatal development, and thereafter undergo somatic segregation during adult life (**Elson et al., 2001**). It is possible that the limited number of cells and cell divisions in some metazoans, like *D. melanogaster*, is insufficient to create the clonal expansion of mtDNA mutations and mosaic respiratory chain deficiency in the adult organism to limit life span. In contrast, somatic mtDNA mutations and mosaic respiratory chain deficiency seem to be ubiquitous findings in old mammals, including humans, and were experimentally linked to the ageing process through studies in the mtDNA mutator mouse strains (**Larsson, 2010**; **Trifunovic et al., 2004**; **Kujoth et al., 2005**). Similar to ageing humans, most somatic mtDNA mutations in the mtDNA mutator mouse are formed during embryogenesis (**Trifunovic et al., 2005**) and clonal expansion of mutated mtDNA has been reported in some organs, e.g., heart (**Trifunovic et al., 2004**) and colonic crypts (**Greaves et al., 2011**). Importantly, the mtDNA mutator mouse model not only clarified the direct link between mtDNA mutations and ageing phenotypes, but it is also a valuable tool to develop strategies to mitigate the ageing process. For example, increasing mitochondrial mass by overexpressing the peroxisome proliferator-activated receptor γ coactivator-1α (PGC-1α), an important regulator of mitochondrial biogenesis, ameliorated the skeletal muscle and heart phenotype of the mtDNA mutator mouse (**Dillon et al., 2012**). However, PGC-1α is also involved in numerous other cellular processes, and aberrant activation can be deleterious, emphasizing the need for alternative strategies to treat the natural ageing process (**Miura et al., 2006**; **Ciron et al., 2012**; **Filograna et al., 2021**). A more direct approach to improve the phenotypes of the mtDNA mutator mouse is to increase the absolute amount of mtDNA molecules by upregulation of the total mtDNA copy number. While this does not affect the fraction of mutated mtDNA, the increase in the absolute number of wild-type mtDNA segments raises the likelihood that sufficient functional gene products can be made to enable sufficient OXPHOS function (**Nishiyama et al., 2010**; **Jiang et al., 2017**). Modulation of mtDNA copy number can be achieved by genetically altering the levels of the mitochondrial transcription factor A (TFAM), which packages mtDNA into nucleoids. Heterozygous knockout of *Tfam* in wild-type mice results in a ~50% decrease in mtDNA levels, whereas moderate overexpression of *Tfam* leads to a ~50% increase in mtDNA levels (**Larsson et al., 1998**; **Bonekamp et al., 2021**). Importantly, moderate decrease or increase in TFAM levels mainly affects mtDNA copy number but does only mildly or not at all alter mtDNA gene expression, respiratory chain function, or mitochondrial mass in mice without mtDNA mutations. Employing this strategy to increase mtDNA levels in the mtDNA mutator mouse revealed a rescuing effect on the early-onset male infertility phenotype at 4 months of age (**Jiang et al., 2017**). Similarly, a beneficial impact was also evident in the m.C5024T tRNA$^{Ala}$ mouse model, where moderate overexpression of TFAM and the associated mtDNA increase improved the cardiomyopathy phenotype in aged mice (**Filograna et al., 2019**). Despite these largely positive outcomes of upregulating mtDNA copy number via moderate TFAM overexpression, care must be taken as strong overexpression of TFAM is known to lead to respiratory

chain deficiency in skeletal muscle and shorten the life span in mice (*Bonekamp et al., 2021*). It is now clear that the TFAM-to-mtDNA ratio determines the compaction of the nucleoid, and in vivo and in vitro experiments have established that a high degree of nucleoid compaction will shut off mtDNA expression (*Bonekamp et al., 2021*; *Farge et al., 2014*; *Brüser et al., 2021*).

Here, we investigated whether mtDNA copy number modulation via moderate TFAM alteration impacts the ageing phenotypes in mtDNA mutator mice. Surprisingly, a moderate increase in TFAM levels did not translate to a proportional modulation of the mtDNA copy number, as previously observed in mice with wild-type mtDNA or in the tRNA$^{Ala}$ mouse model. Instead, the mtDNA mutator mouse shows a trend towards endogenous upregulation of mtDNA copy number in some tissues, and experimental modulation of TFAM will be added to this increase to result in tissue-specific effects with varying TFAM-to-mtDNA ratios that impact mtDNA gene expression in different ways. In colon, TFAM overexpression reduced mtDNA gene expression, negatively affecting OXPHOS and leading to a marked increase in expression of *Methylenetetrahydrofolate dehydrogenase 2* (*Mthfd2*), a marker for mitochondrial dysfunction. In contrast, in brown adipose tissue (BAT), a decrease in TFAM levels normalized *Uncoupling protein 1* (*Ucp1*) expression. In summary, mtDNA copy number regulation is more complex than suggested by previous studies (*Nishiyama et al., 2010*; *Jiang et al., 2017*; *Larsson et al., 1998*; *Bonekamp et al., 2021*; *Filograna et al., 2019*) and the TFAM-to-mtDNA ratio seems to be finely tuned in a tissue-specific manner. Alterations of the ratio can have different outcomes in different tissues and attempts to increase TFAM levels to counteract pathologies caused by mitochondrial dysfunction must, therefore, be carefully evaluated.

## Results

### Moderate alterations of TFAM levels only mildly affect pathology in aged mtDNA mutator mice

To experimentally test the impact of mtDNA copy number modulation on the premature ageing phenotypes of mtDNA mutator mice, we generated hemizygous Polg$^{-/mut}$ mice with moderately increased (*Tfam$^{+/OE}$*) or decreased (*Tfam$^{+/-}$*) TFAM levels. We first created heterozygous Polg knockout (*Polg$^{+/-}$*) females that also carry either a *Tfam$^{+/OE}$* or *Tfam$^{+/-}$* allele, and subsequently mated those females to heterozygous mtDNA mutator (*Polg$^{+/mut}$*) males (*Figure 2—figure supplement 1*). Employing this mating scheme ensures that the resulting mice (*Polg$^{-/mut}$*) carry a substantial load of de novo generated somatic mtDNA mutations and prevent them from inheriting maternally transmitted mtDNA mutations from their mothers, which otherwise would aggravate the phenotype (*Ross et al., 2013*). Therefore, in the studied *Polg$^{-/mut}$* mice, all mtDNA mutations are generated de novo during embryonic and postnatal life. Both the 'classical' mtDNA mutator mice (*Polg$^{mut/mut}$*) (*Trifunovic et al., 2004*; *Kujoth et al., 2005*) and the mtDNA mutator mice generated in this study (*Polg$^{-/mut}$*) develop profound premature ageing phenotypes at the age of 35 weeks. The mtDNA mutator mice (*Polg$^{-/mut}$*) had a markedly reduced body weight at the age of 35 weeks and increasing TFAM levels (*Polg$^{-/mut}$*; *Tfam$^{+/OE}$*) led to a statistically significant further, albeit slight, reduction of body weight (*Figure 1A*). In contrast, the reduction of TFAM expression (*Polg$^{-/mut}$*; *Tfam$^{+/-}$*) did not have any statistically significant effect on the reduced body weight of mtDNA mutator mice (*Figure 1A*). The organ-to-body weight ratios were largely unaffected by variations in TFAM expression, with the exception of testis. In testis, we observed a clear rescue effect in mtDNA mutator mice with increased TFAM levels (*Polg$^{-/mut}$*; *Tfam$^{+/OE}$*) resulting in a testis to body weight ratio similar to wild-type mice, whereas reduced TFAM levels (*Polg$^{-/mut}$*; *Tfam$^{+/-}$*) exacerbated the phenotype (*Figure 1B-D*), consistent with our previous study (*Jiang et al., 2017*). These results demonstrate that TFAM modulation does not grossly impact most ageing phenotypes of the mtDNA mutator mouse.

### Tissue-specific effects on mtDNA copy number and TFAM-to-mtDNA ratios

The beneficial effect of moderate TFAM overexpression on the testis phenotype, and the absence of any rescue effect on the increased heart and spleen phenotype of *Polg$^{-/mut}$*; *Tfam$^{+/OE}$* mice prompted us to perform further characterization of mtDNA expression in different tissues. To this end, we analyzed the relative mtDNA copy number by qPCR using three different probes, i.e., *NADH-ubiquinone oxidoreductase chain 1* (*mt-Nd1*), *ATP synthase subunit a* (*mt-Atp6*), and *Cytochrome b* (*Mt-CytB*)

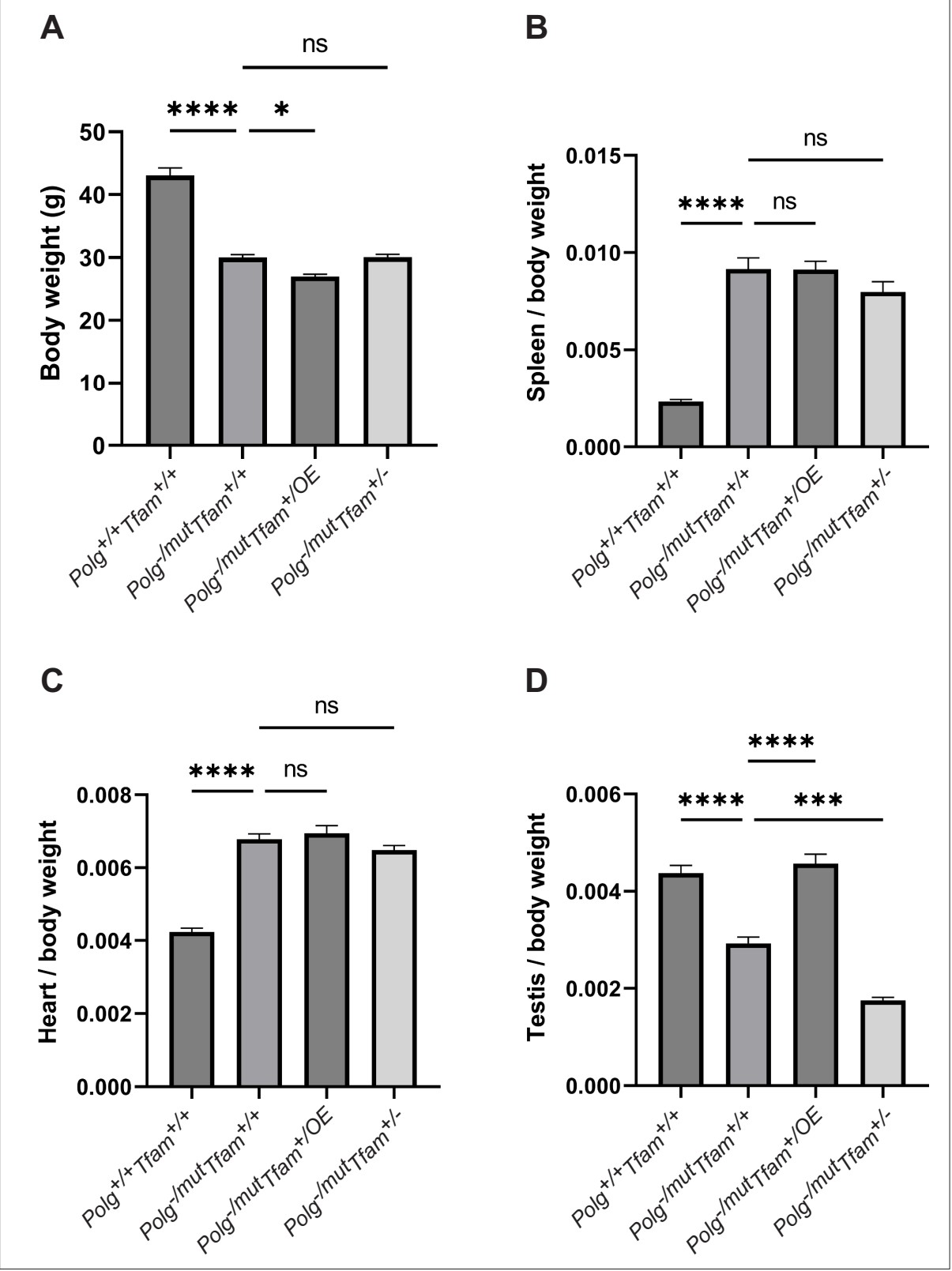

**Figure 1.** Mitochondrial transcription factor A (TFAM) modulation does not rescue the decreased body weight or the increased spleen/body weight and heart/body weight ratio. (**A**) Body weight in g. (**B**) Spleen to body weight ratio (g/g). (**C**) Heart to body weight ratio (g/g). (**D**) Testis to body weight ratio (g/g). n≥7. Data are represented as mean ± SEM; *p<0.05; **p<0.01; ***p<0.001; ns: non-significant.

*Figure 1 continued on next page*

Figure 1 continued

The online version of this article includes the following figure supplement(s) for figure 1:

**Figure supplement 1.** Mating strategy to generate mitochondrial DNA (mtDNA) mutator mice in combination with the Tfam[+/OE] or Tfam[+/-] alleles.

(*Figure 2*). It should be noted that the mtDNA mutator mice carry linear deleted mtDNA molecules spanning the major arc of mtDNA (*Trifunovic et al., 2004*; *Figure 2A*). These linear deletions are continuously formed replication intermediates and represent around 30% of the total mtDNA in all investigated tissues (*Trifunovic et al., 2004*; *Bailey et al., 2009*). *Mt-Nd1* hybridizes to a genomic region corresponding to the minor arc of the mtDNA and is hence a good proxy for full-length mtDNA, whereas *mt-Atp6* and *mt-CytB* hybridize to genomic regions in the major arc of mtDNA and, therefore, are good indicators of total mtDNA levels, including deleted mtDNA molecules (*Figure 2A*). The mtDNA mutator mice spontaneously upregulated the total mtDNA copy number in several tissues (*Figure 2B-F*, *Figure 2—figure supplement 1A*), similar to what has previously been shown for mice harboring a point mutation in the tRNA[Ala] gene (*Filograna et al., 2019*). Surprisingly, increased TFAM levels did not cause any additional relative mtDNA copy number change in either liver or heart of *Polg*[-/mut]; *Tfam*[+/OE] mice and only a very subtle difference in the colon (*Figure 2B–D*). The qPCR results for the liver were independently confirmed by Southern blot analyses (*Figure 2—figure supplement 1B and C*). In contrast, increased TFAM levels caused an additional relative mtDNA copy number increase in brown adipose tissue and spleen of *Polg*[-/mut]; *Tfam*[+/OE] mice compared to mtDNA mutator mice (*Figure 2E and F*).

Changes in TFAM levels without a concomitant change in mtDNA levels are potentially problematic, because the TFAM-to-mtDNA ratio will be impacted to affect nucleoid compaction, which, in turn, impacts mtDNA expression (*Bonekamp et al., 2021*; *Farge et al., 2014*; *Brüser et al., 2021*). We, therefore, proceeded to calculate the relative TFAM-to-mtDNA ratios in liver, heart, colon, spleen, and brown adipose tissue. Compared to wild-type mice, *Polg*[-/mut]; *Tfam*[+/+] mice had a moderately elevated TFAM-to-mtDNA ratio in the heart and a drastically decreased ratio in the spleen. TFAM overexpression in the mtDNA mutator mice (*Polg*[-/mut]; *Tfam*[+/OE] mice) resulted in a marked additional increase of the TFAM-to-mtDNA ratio in the liver and an additional increase in the heart in comparison with *Polg*[-/mut]; *Tfam*[+/+] mice, whereas no major effect was observed in the other tissues (*Table 1*). When the TFAM levels were instead decreased (*Polg*[-/mut]; *Tfam*[+/-] mice), the TFAM-to-mtDNA ratios were decreased in liver, colon, and BAT in comparison with *Polg*[-/mut]; *Tfam*[+/+] mice, whereas no additional change was seen in heart (*Table 1*).

Taken together, these findings show that modulation of TFAM expression in the severely affected mtDNA mutator mouse has tissue-specific effects on TFAM-to-mtDNA ratios, which is well known to influence nucleoid compaction (*Bonekamp et al., 2021*). We, therefore, proceeded to investigate the pathophysiological effects of TFAM modulation in different organs of the mtDNA mutator mouse. We hypothesized that in tissues where TFAM overexpression increases the TFAM-to-mtDNA ratio, the mtDNA expression will be reduced and negatively impact tissue function. In contrast, we speculated that in tissues where elevated TFAM levels increase mtDNA copy number, the mitochondrial function would improve, in line with previous results (*Jiang et al., 2017*; *Filograna et al., 2019*).

## Moderate TFAM overexpression impairs mtDNA expression in the liver of mtDNA mutator mice

In the liver, TFAM overexpression resulted in a decrease of *mt-Atp6* and *mt-Cytb* transcript levels (*Figure 3A*) as predicted by the increased TFAM-to-mtDNA ratio. The decrease in gene expression was also evident on the protein level as the steady-state levels of several subunits of the OXPHOS complexes were further decreased when compared to *Polg*[-/mut]; *Tfam*[+/+] mice (*Figure 3B*). Of note, we did not only observe a reduction in levels of proteins encoded by the mtDNA, e.g., Cytochrome c oxidase subunit 1 (COX1) and Cytochrome c oxidase subunit 2 (COX2) of complex IV, but also of OXPHOS subunits encoded by the nuclear DNA, e.g., NADH:Ubiquinone Oxidoreductase Subunit B8 (NDUFB8) of complex I. This is an expected secondary effect as mtDNA gene expression is necessary for the stability of nucleus-encoded OXPHOS complex subunits (*Kühl et al., 2017*). The levels of the ATP Synthase F1 Subunit Alpha (ATP5A) protein of complex V (ATP synthase) were largely unaffected, which goes well in line with previous reports showing that impaired mtDNA expression leads to the

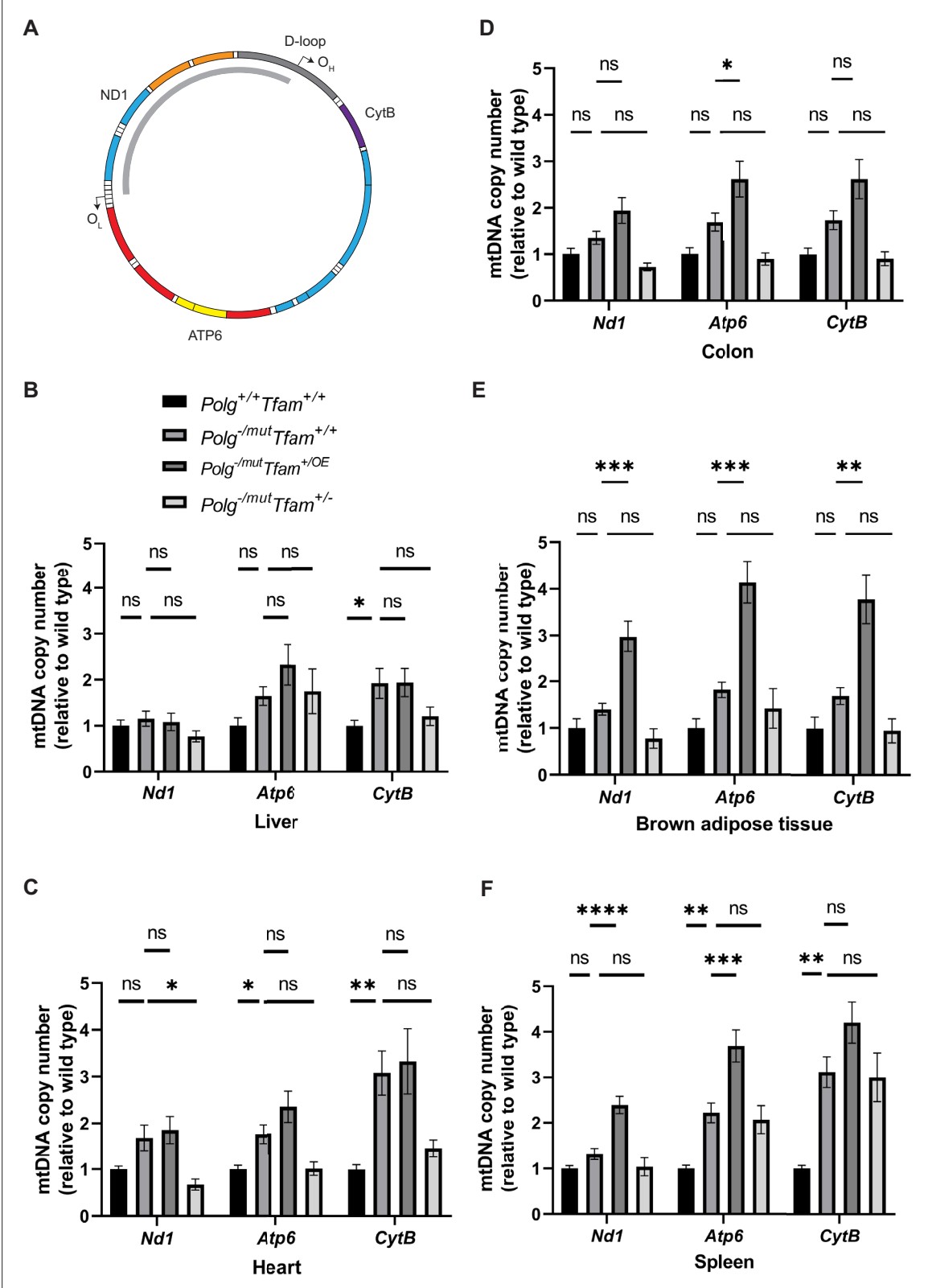

**Figure 2.** Modulation of mitochondrial transcription factor A (TFAM) expression affects mitochondrial DNA (mtDNA) copy number in a tissue-specific manner. (**A**) Schematic of the mtDNA highlighting the position of the probes used for mtDNA copy number analysis by qPCR. The deleted mtDNA region is indicated with a gray arc. (**B–F**) Relative mtDNA copy number quantification (*Nd1*/18 S, *Atp6*/18 S, *Cytb*/18 S) in (**B**) liver, (**C**) heart, (**D**) colon, (**E**) brown adipose tissue, and (**F**) spleen. n≥5. Data are represented as mean ± SEM; *p<0.05; **p<0.01; ***p<0.001; ns: non-significant.

*Figure 2 continued on next page*

*Figure 2 continued*

The online version of this article includes the following source data and figure supplement(s) for figure 2:

**Source data 1.** Excel file containing qPCR data for *Figure 2B–F*.

**Figure supplement 1.** Quantification of mitochondrial DNA (mtDNA) levels by qPCR in testis and by Southern blot analysis in liver.

**Figure supplement 1—source data 1.** Excel file containing qPCR data for *Figure 2—figure supplement 1A*.

**Figure supplement 1—source data 2.** Original files of mitochondrial DNA (mtDNA) Southern blot analysis for *Figure 2—figure supplement 1B*.

**Figure supplement 1—source data 3.** Original files of 18 S rDNA Southern blot analysis for *Figure 2—figure supplement 1B*.

**Figure supplement 1—source data 4.** PDF file containing Southern blot analysis for *Figure 2—figure supplement 1B*, indicating relevant bands.

formation of a stable subcomplex containing the F1 subunit of complex V of the OXPHOS system (*Kühl et al., 2017*). We detected increased levels of the complex II subunit Succinate Dehydrogenase Complex Iron Sulfur Subunit B (SDHB). Complex II is exclusively nuclear encoded and a compensatory increase upon impaired mitochondrial gene expression has been observed before (*Kühl et al., 2017*).

We proceeded to measure the enzyme activities of individual OXPHOS complexes in liver mitochondria (*Figure 3C*). The complex I and complex IV activities were reduced to about 50% in *Polg*$^{-/mut}$; *Tfam*$^{+/+}$ mice in comparison with wild-type mice (*Figure 3C*). However, we did not see any further alteration of the reduced enzyme activities induced by TFAM overexpression or reduced TFAM expression (*Figure 3C*). Interestingly, we detected a significant increase in complex II and complex II + complex III activity upon TFAM overexpression, which can partially be explained by the increased complex II protein levels we observed in *Polg*$^{-/mut}$; *Tfam*$^{+/OE}$ mice (*Figure 3B and C*). The *Polg*$^{-/mut}$; *Tfam*$^{+/+}$ mice had normal expression of *Activating Transcription Factor 4* (*Atf4*) and *Activating Transcription Factor 5* (*Atf5*) in liver, whereas the expression of *Mthfd2* was markedly increased (*Figure 3D*). We have previously reported that increased *Mthfd2* expression is a sensitive marker of mitochondrial dysfunction and that *Mthfd2* expression increases as OXPHOS dysfunction progresses (*Kühl et al., 2017*). The finding of increased *Mthfd2* expression in *Polg*$^{-/mut}$; *Tfam*$^{+/+}$ liver is thus consistent with the observed OXPHOS dysfunction (*Figure 3C and D*). This was unaltered by TFAM overexpression.

We also observed an increase in Fibroblast Growth Factor 21 (FGF21) levels in plasma of *Polg*$^{-/mut}$; *Tfam*$^{+/+}$ mice pointing to a negative effect on energy homeostasis (*Figure 3E*). Interestingly, the plasma levels of FGF21 were further increased in *Polg*$^{-/mut}$; *Tfam*$^{+/OE}$ mice (*Figure 3E*). These findings are consistent with the proposed action of FGF21 as a starvation hormone and may reflect the progression of body weight loss in *Polg*$^{-/mut}$; *Tfam*$^{+/+}$ and additional body weight reduction in *Polg*$^{-/mut}$; *Tfam*$^{+/OE}$ mice (*Figure 1A*).

In summary, moderate TFAM overexpression does not improve the liver phenotype of mtDNA mutator mice. On the contrary, elevating TFAM had a negative effect on mtDNA gene expression (*Figure 3A and B*), which could be linked to the increased TFAM-to-mtDNA ratio (*Table 1*) causing tighter compaction of the mitochondrial nucleoid. The mtDNA mutator mouse has abundant point mutations in mtDNA that will affect tRNAs and rRNAs, thus impairing mitochondrial translation, as well as abundant point mutations causing amino acid substitutions in the protein-coding genes of mtDNA, thus causing dysfunction or impaired stability of respiratory chain complexes. The impaired mtDNA gene expression caused by increased nucleoid compaction poses an additional burden on the already compromised OXPHOS dysfunction in mtDNA mutator mice. The marked additional increase

**Table 1.** Relative mitochondrial transcription factor A (TFAM)-to-mitochondrial DNA (mtDNA) ratios in different tissues.
The TFAM-to-mtDNA ratio was calculated from normalized TFAM protein levels (n=2) and normalized mtDNA levels (n=5) as determined by the *mt-Nd1* probe. The respective values for the normalized TFAM and mtDNA levels are indicated in parentheses. BAT: brown adipose tissue.

|  | *Polg*$^{+/+}$*Tfam*$^{+/+}$ | *Polg*$^{-/mut}$*Tfam*$^{+/+}$ | *Polg*$^{-/mut}$*Tfam*$^{+/OE}$ | *Polg*$^{-/mut}$ *Tfam*$^{+/-}$ |
|---|---|---|---|---|
| Liver | 1.00 | 0.86 (0.99:1.15) | 2.14 (2.32:1.08) | 0.48 (0.37:0.77) |
| Heart | 1.00 | 1.31 (2.21:1.69) | 1.93 (3.59:1.86) | 1.24 (0.83:0.67) |
| Colon | 1.00 | 1.19 (1.62:1.35) | 1.01 (1.96:1.95) | 0.61 (0.44:0.73) |
| BAT | 1.00 | 0.95 (1.33:1.40) | 1.09 (3.23:2.97) | 0.67 (0.52:0.78) |
| Spleen | 1.00 | 0.32 (0.42:1.32) | 0.28 (0.66:2.39) | 0.19 (0.20:1.04) |

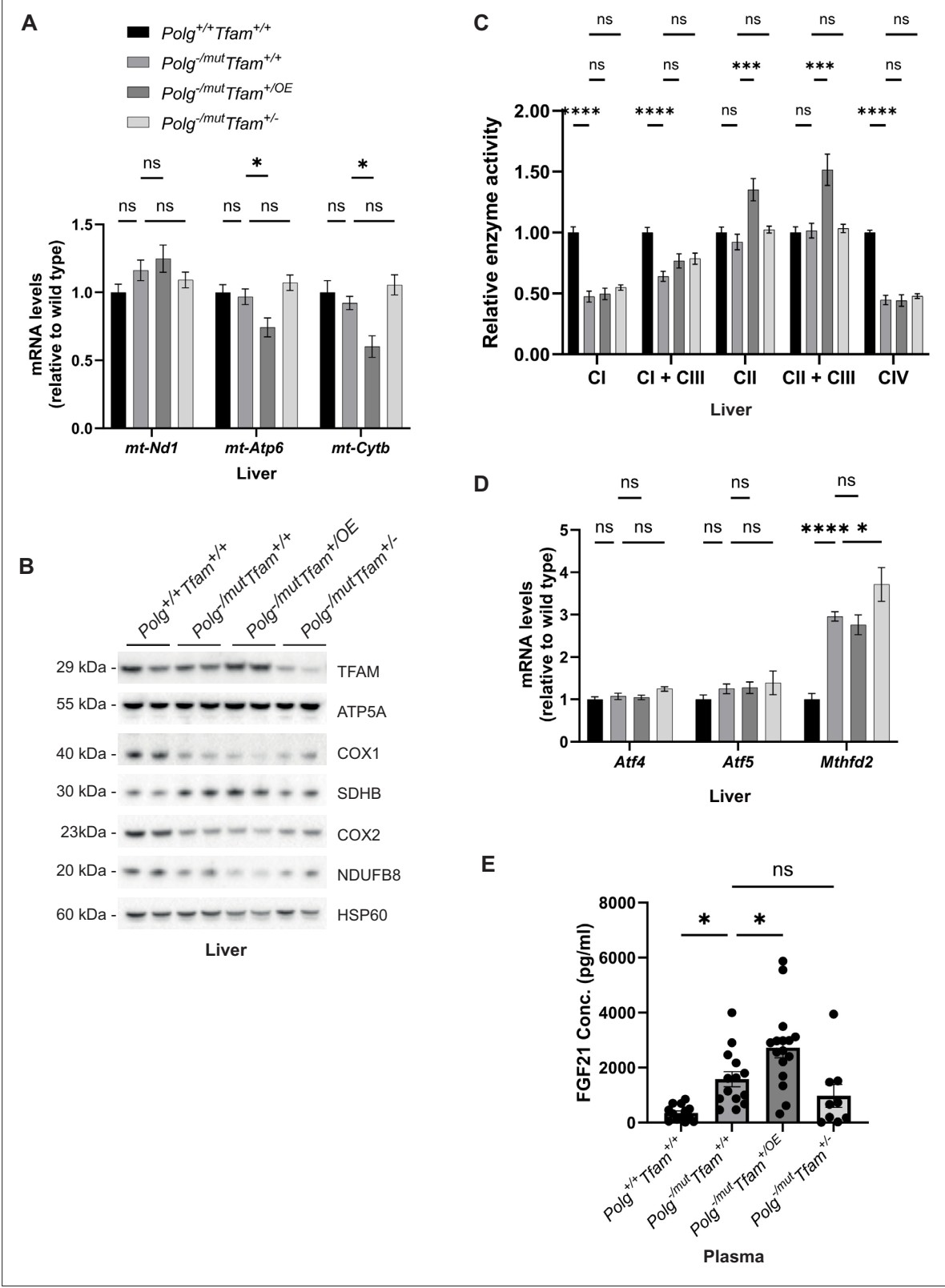

**Figure 3.** Moderate mitochondrial transcription factor A (TFAM) overexpression negatively impacts mitochondrial DNA (mtDNA) gene expression and tissue physiology in liver and correlates with increased fibroblast growth factor 21 (FGF21) levels. (**A**) Relative expression levels of mtDNA-encoded transcripts (*Nd1*/β-*actin*, *Atp6*/β-*actin*, *Cytb*/β-*actin*) measured by RT-qPCR in liver. n≥7. Data are represented as mean ± SEM; *p<0.05; **p<0.01; ***p<0.001; ns: non-significant. (**B**) Western blot analysis of steady-state levels of mitochondrial proteins in liver. (**C**) Relative enzyme activities of

*Figure 3 continued on next page*

*Figure 3 continued*

oxidative phosphorylation (OXPHOS) complexes measured by spectrophotometry in liver mitochondria. n≥6 Data are represented as means ± SEM; *p<0.05; **p<0.01; **p<0.01; ***p<0.001; ns: non-significant. (**D**) Relative expression levels of mitochondrial stress markers (*Atf4/β-actin*, *Atf5/β-actin*, *Mthfd2/β-actin*) measured by RT-qPCR in liver. n≥7. Data are represented as means ± SEM; *p<0.05; **p<0.01; ***p<0.001, ns: non-significant (**E**) Quantification of FGF21 levels in plasma measured by ELISA. n ≥ 9. Data are represented as means ± SEM; *p<0.05; **p<0.01; ***p<0.001; ns: non-significant.

The online version of this article includes the following source data for figure 3:

**Source data 1.** Excel file containing qPCR data for *Figure 3A, D and E*.

**Source data 2.** Original files of Western blot analysis for *Figure 3B*.

**Source data 3.** PDF file containing Western blot analysis for *Figure 3B*, indicating relevant bands.

in FGF21 levels in *Polg^-/mut*; *Tfam^+/OE* mice indicates that TFAM overexpression has a negative impact on liver physiology in mtDNA mutator mice.

## Modulation of TFAM does not impact the cardiomyopathy phenotype of mtDNA mutator mice

In the heart of *Polg^-/mut*; *Tfam^+/+* mice, we detected strongly reduced levels of mtDNA-encoded transcripts (*Figure 4A*). This can potentially be explained by the substantial increase in TFAM protein levels in these animals (*Figure 4B*), which, given the absence of a corresponding increase in full-length mtDNA, results in a higher TFAM-to-mtDNA ratio and thus probably causes reduced mtDNA expression (*Table 1*). Importantly, TFAM modulation did not have any effect on the reduced transcript levels or the diminished protein levels of OXPHOS complexes in mtDNA mutator mice (*Figure 4A and B*).

Consistent with the increased heart-to-body-weight ratio of *Polg^-/mut*; *Tfam^+/+* mice (*Figure 1C*), the expression of *Natriuretic Peptide A* (*Nppa*), a marker for heart failure, and *Mthfd2*, as described above a marker for OXPHOS dysfunction, were increased, whereas there was no change in mRNA levels for *Atf4* and *Atf5* (*Figure 4C*). The levels of *Atf4*, *Atf5*, *Mthfd2*, and *Nppa* mRNAs were not impacted by reduced or increased expression of TFAM (*Figure 4C*). These findings are in agreement with the observation that TFAM overexpression does not rescue the increased heart-to-body weight ratio observed in mtDNA mutator mice (*Figure 1C*).

All in all, these results demonstrate that the mtDNA mutator hearts upregulate TFAM protein levels, likely as a compensatory mechanism. However, the increased TFAM levels do not lead to a concomitant increase in mtDNA levels, but instead result in an increased TFAM-to-mtDNA ratio and reduced steady-state levels of mitochondrial transcripts. TFAM overexpression does not further affect this endogenous compensatory response and causes no additional increase in mtDNA levels. Thus, neither moderate TFAM overexpression nor reduced TFAM expression affect the heart phenotype of mtDNA mutator mice.

## Increased TFAM levels do not rescue OXPHOS dysfunction in the colon of mtDNA mutator mice

In the colon, we found an increase in levels of mtDNA-encoded mRNAs in *Polg^-/mut*; *Tfam^+/+* mice (*Figure 5A*). This response was largely ablated upon moderate TFAM overexpression. In line with this, levels of key OXPHOS subunits were further reduced in *Polg^-/mut*; *Tfam^+/OE* mice in comparison to *Polg^-/mut*; *Tfam^+/+* mice (*Figure 5B*). However, this did not lead to a further deterioration of levels of respiratory chain complexes on blue-native PAGE gels (*Figure 5C*) or respiratory chain enzyme activities (*Figure 5D*), possibly due to their already extremely low levels in *Polg^-/mut*; *Tfam^+/+* mice. We found a strong induction of the *Atf5* and *Mthfd2* mRNAs in the colon of *Polg^-/mut*; *Tfam^+/+* mice. Upon TFAM overexpression, *Mthfd2* mRNA expression levels were further increased (*Figure 5E*).

In summary, the responses in the colon have similarities to the responses in the liver as TFAM overexpression results in a reduction of mtDNA-encoded transcripts in mtDNA mutator mice. This reduction likely adds to the already drastic decrease in protein levels of several subunits of the OXPHOS complexes. The strongly induced expression of *Mthfd2* suggests that instead of having a beneficial effect, TFAM overexpression results in a deterioration of colon physiology.

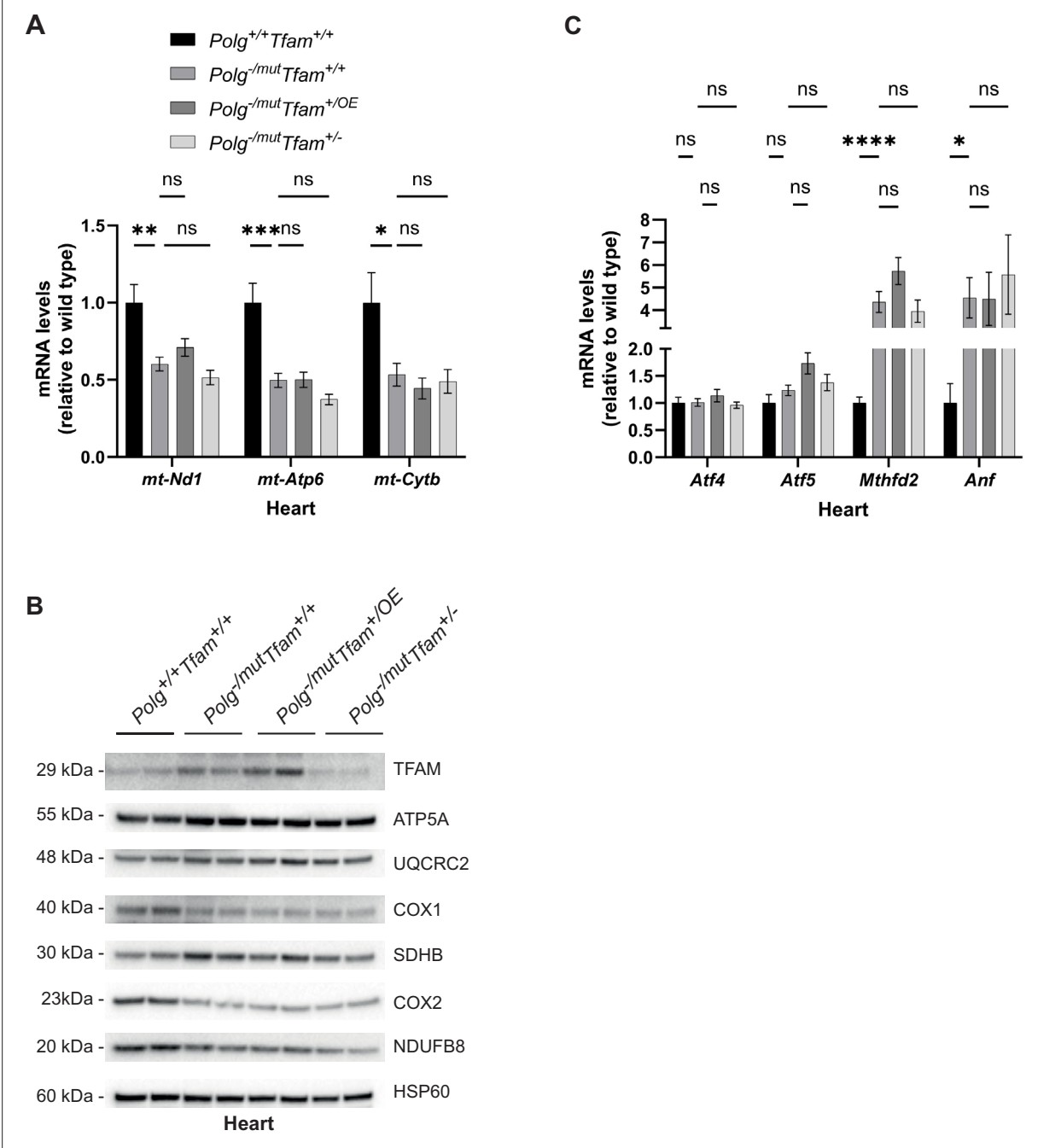

**Figure 4.** Alteration of mitochondrial transcription factor A (TFAM) expression does not affect the heart phenotype of mitochondrial DNA (mtDNA) mutator mice. (**A**) Relative expression levels of mtDNA-encoded transcripts (*Nd1/β-actin, Atp6/β-actin, Cytb/β-actin*) measured by RT-qPCR in heart. n ≥ 7. Data are represented as mean ± SEM; *p<0.05; **p<0.01; ***p<0.001; ns: non-significant. (**B**) Western blot analysis of steady-state levels of mitochondrial proteins in heart. (**C**) Relative expression levels of mitochondrial stress markers (*Atf4/β-actin, Atf5/β-actin, Mthfd2/β-actin, Nppa/β-actin*) measured by RT-qPCR in heart. n ≥ 7. Data are represented as mean ± SEM; *p<0.05; **p<0.01; ***p<0.001; ns: non-significant.

The online version of this article includes the following source data for figure 4:

Source data 1. Excel file containing qPCR data for *Figure 4A and C*.

Source data 2. Original files of Western blot analysis for *Figure 4B*.

Source data 3. PDF file containing Western blot analysis for *Figure 4B*, indicating relevant bands.

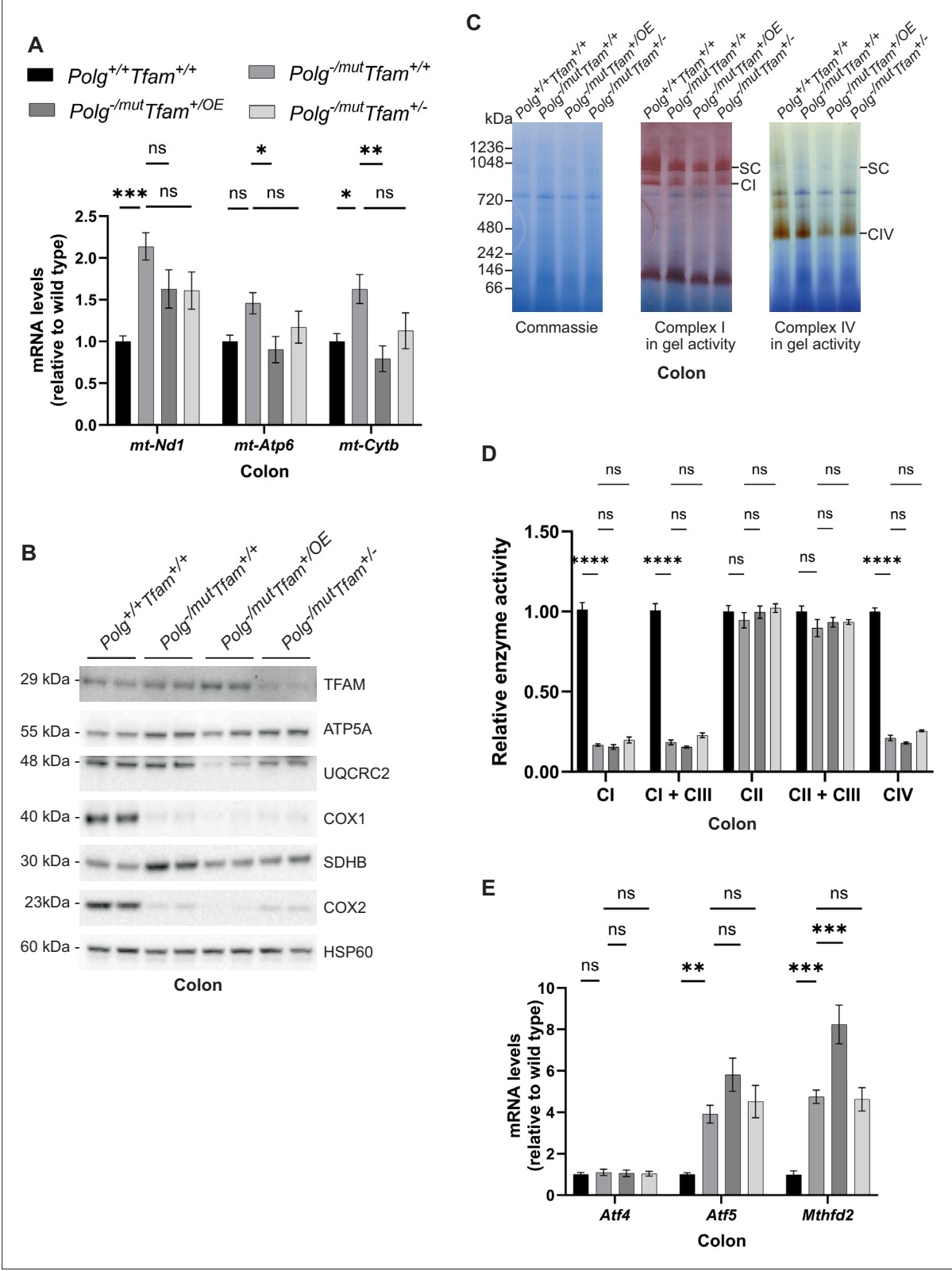

**Figure 5.** Increased mitochondrial transcription factor A (TFAM) levels do not rescue the reduced oxidative phosphorylation (OXPHOS) function in the colon of mitochondrial DNA (mtDNA) mutator mice. (**A**) Relative expression levels of mtDNA-encoded transcripts (*Nd1*/β-*actin*, *Atp6*/β-*actin*, *Cytb*/β-*actin*) measured by RT-qPCR in colon. n≥7. Data are represented as mean ± SEM; *p<0.05; **p<0.01; ***p<0.001; ns: non-significant. (**B**) Western blot analysis of steady-state levels of mitochondrial proteins in colon. (**C**) BN-PAGE and in-gel activities of complex I and complex IV activities in

*Figure 5 continued on next page*

*Figure 5 continued*

mitochondrial protein extracts from mouse colon. Coomassie staining of the gel is shown to indicate equal loading. SC, Supercomplexes. (**D**) Relative enzyme activities of OXPHOS complexes measured by spectrophotometry in colon mitochondria. n≥3. Data are represented as mean ± SEM; *p<0.05; **p<0.01; ***p<0.001; ns: non-significant. (**E**) Relative expression levels of mitochondrial stress markers (*Atf4/β-actin*, *Atf5/β-actin*, *Mthfd2/β-actin*) measured by RT-qPCR in colon. n≥9. Data are represented as mean ± SEM; *p<0.05; **p<0.01; ***p<0.001; ns: non-significant.

The online version of this article includes the following source data for figure 5:

**Source data 1.** Excel file containing qPCR data for *Figure 5A and E*.

**Source data 2.** Original files of Western blot analysis for *Figure 5B*.

**Source data 3.** Original files of BN-PAGE analysis for *Figure 5C*.

**Source data 4.** PDF file containing Western blot analysis for *Figure 5B*, indicating relevant bands.

**Source data 5.** PDF file containing BN-PAGE analysis for *Figure 5C*, indicating relevant bands.

## TFAM downregulation rescues Ucp1 expression in brown adipose tissue of mtDNA mutator mice

TFAM overexpression in BAT resulted in a significant increase in mtDNA copy number and did not alter the TFAM-to-mtDNA ratio (*Figure 2E* and *Table 1*). In line with this finding, the levels of mtDNA-encoded transcripts (*Figure 6A*) and OXPHOS subunits (*Figure 6B*) did not change. In contrast, the reduced TFAM expression in *Polg*[-/mut]; *Tfam*[+/-] mice led to increased steady-state levels of mtDNA-encoded transcripts (*Figure 6A*), which correlate well with the decrease in TFAM-to-mtDNA ratios in BAT of these mice. The levels of OXPHOS subunits in *Polg*[-/mut]; *Tfam*[+/-] mice were only mildly increased or not changed (*Figure 6B*). UCP1 protein levels are reduced in mtDNA mutator mice and TFAM overexpression did not influence the UCP1 levels in BAT. Surprisingly, the reduced TFAM expression in *Polg*[-/mut]; *Tfam*[+/-] mice resulted in elevated UCP1 protein levels (*Figure 6B*). Consistent with this result, the expression of *Ucp1* and *Cell Death Inducing DFFA Like Effector A* (*Cidea*) mRNAs, which are markers for the thermogenic competence of BAT, was strongly induced in *Polg*[-/mut]; *Tfam*[+/-] mice compared to mtDNA mutator mice (*Figure 6C*). These findings argue that the restoration of mtDNA expression may be important for maintaining the differentiated state of BAT in *Polg*[-/mut] mice or that homeostatic mechanisms induced by altered function of other tissues affect nuclear gene expression in BAT.

Taken together, our data show that moderate TFAM overexpression in BAT causes an increase in mtDNA copy number without affecting mtDNA expression. While this could potentially lead to a beneficial effect on BAT, we did not observe any rescue effect when assessing transcript and protein levels of key factors important for BAT function. In contrast, reduced TFAM levels led to a considerable burst in mtDNA gene expression, which surprisingly ameliorated the expression of markers for the thermogenic competence of BAT, hence likely positively affecting BAT function.

## Upregulation of TFAM does not impact mitochondrial gene expression in mtDNA mutator spleen but impacts cytokine levels

In the spleen of *Polg*[-/mut]; *Tfam*[+/+] mice, levels of mtDNA-encoded transcripts were substantially increased (*Figure 7A*). This is likely connected to the decrease in TFAM protein levels in these animals (*Figure 7B*), which is not accompanied by a corresponding decrease in mtDNA levels (Figure 2F) but instead results in a markedly decreased TFAM-to-mtDNA ratio (*Table 1*). Despite the elevated transcript levels in the spleen of mtDNA mutator mice, the levels of the complex IV subunit COX2 were drastically decreased (*Figure 7B*). TFAM overexpression, which further increased mtDNA copy number in spleen (*Figure 2F*), did not affect the TFAM-to-mtDNA ratio (*Table 1*) or mtDNA gene expression (*Figure 7A and B*). This resembles the previously reported situation in testis and heart, where moderate TFAM overexpression in young *Polg*[mut/mut] mice or aged tRNA[Ala]-mutant mice, respectively, exerted a beneficial effect by increasing the absolute amount of mtDNA without affecting the mtDNA mutation load (*Jiang et al., 2017*; *Filograna et al., 2019*). To assess whether the increased mtDNA copy number in *Polg*[-/mut]; *Tfam*[+/OE] mice led to amelioration of the premature ageing phenotype in the spleen, we measured the levels of several cytokines in plasma. The mtDNA mutator mice demonstrated elevated Interleukin 5 (IL-5) and C-C Motif Chemokine Ligand 2 (CCL2) cytokine levels in plasma and increased *IL-5* transcript levels in spleen (*Figure 7C and D*). TFAM overexpression

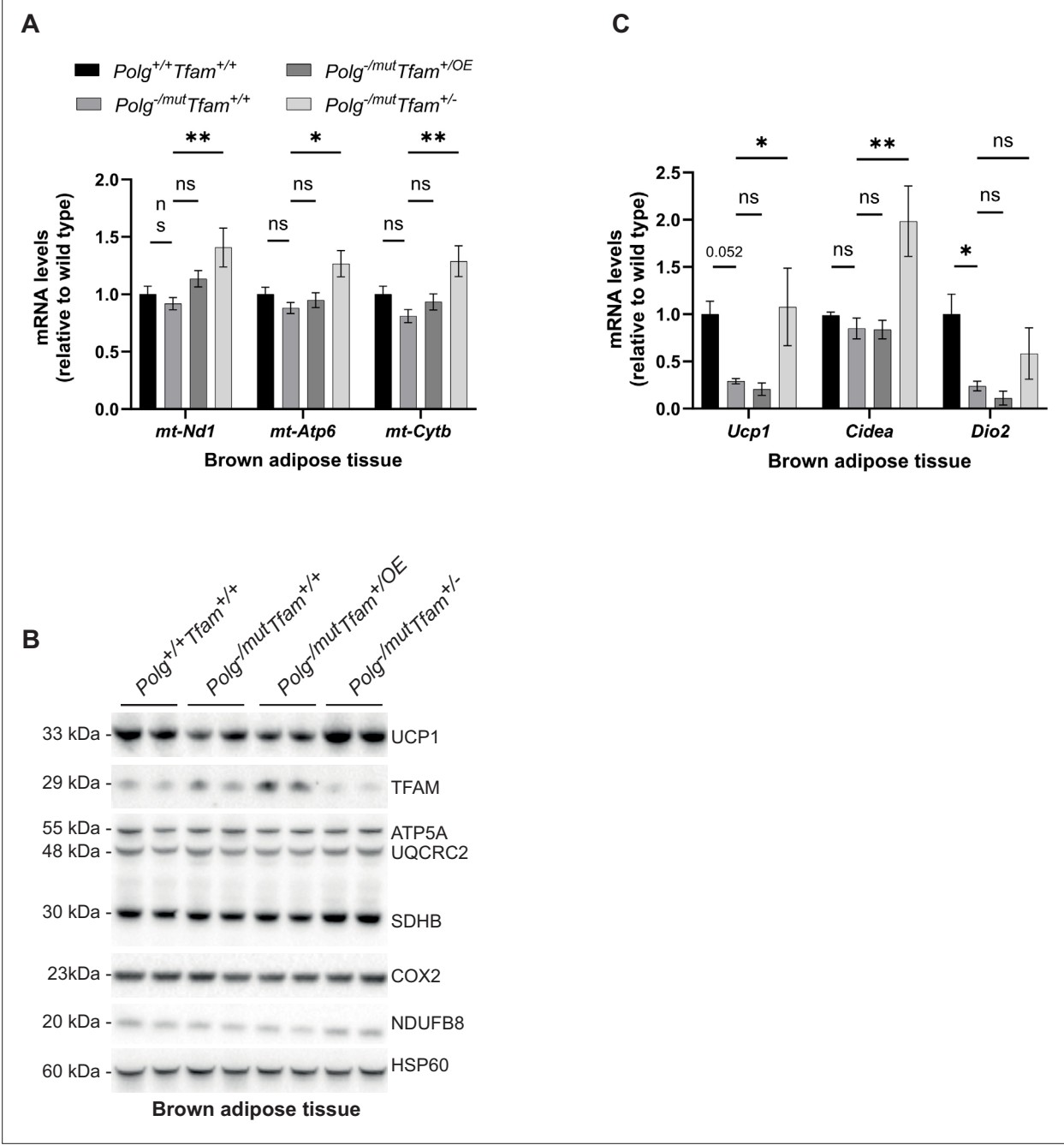

**Figure 6.** Reduction of mitochondrial transcription factor A (TFAM) levels in brown adipose tissue has beneficial effects. (**A**) Relative expression levels of mitochondrial DNA (mtDNA)-encoded transcripts (*Nd1*/β-*actin*, *Atp6*/β-*actin*, *Cytb*/β-*actin*) measured by RT-qPCR in brown adipose tissue (BAT). n≥5. Data are represented as mean ± SEM; *p<0.05; **p<0.01; ***p<0.001; ns: non-significant. (**B**) Western blot analysis of steady-state levels of UCP1 and mitochondrial proteins in BAT. (**C**) Relative expression levels of brown adipose stress markers (*Ucp1*/β-*actin*, *Cidea*/β-*actin*, *Dio2*/β-*actin*) measured by RT-qPCR in BAT. n≥5. Data are represented as mean ± SEM; *p<0.05; **p<0.01; ***p<0.001; ns: non-significant.

The online version of this article includes the following source data for figure 6:

**Source data 1.** Excel file containing qPCR data for *Figure 6A and C*.

**Source data 2.** Original files of Western blot analysis for *Figure 6B*.

**Source data 3.** PDF file containing Western blot analysis for *Figure 6B*, indicating relevant bands.

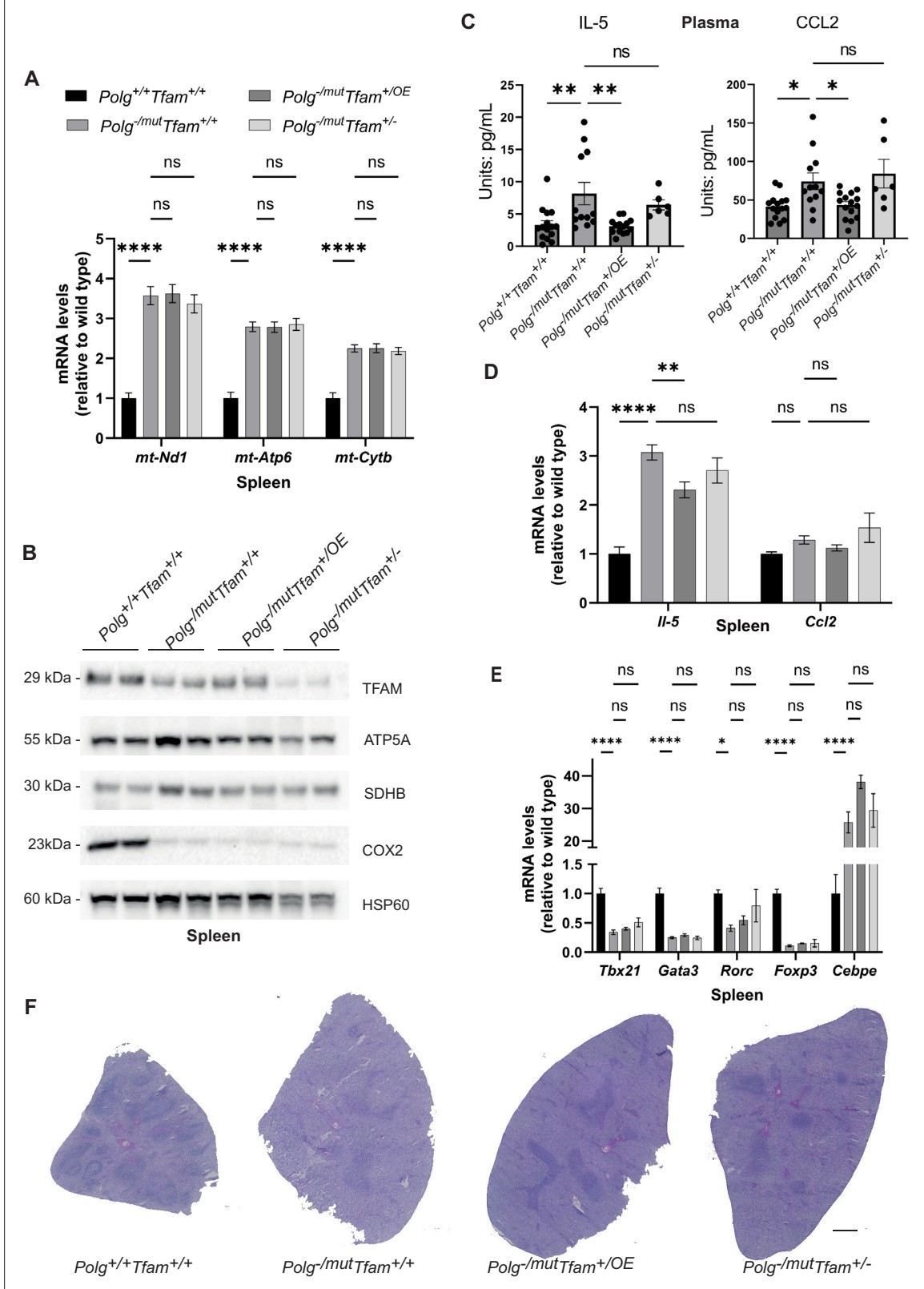

**Figure 7.** Mitochondrial transcription factor A (TFAM) overexpression restores IL-5 and Chemokine Ligand 2 (CCL2) cytokine levels in the spleen of mitochondrial DNA (mtDNA) mutator mice. (**A**) Relative expression levels of mtDNA-encoded transcripts (*Nd1*/β-*actin*, *Atp6*/β-*actin*, *Cytb*/β-*actin*) measured by RT-qPCR in spleen. n≥5. Data are represented as mean ± SEM; *p<0.05; **p<0.01; ***p<0.001; ns: non-significant. (**B**) Western blot analysis of steady-state levels of mitochondrial proteins in spleen. (**C**) Quantification of IL-5 and CCL2 cytokine levels in plasma measured by the

*Figure 7 continued on next page*

*Figure 7 continued*

Mouse Cytokine/Chemokine 44-Plex Discovery Assay. n≥6. Data are represented as mean ± SEM; *p<0.05; **p<0.01; ***p<0.001; ns: non-significant. (**D**) Quantification of *Il-5* and *Ccl2* cytokine transcript levels in spleen measured by RT-qPCR. n≥10. Data are represented as mean ± SEM; *p<0.05; **p<0.01; ***p<0.001; ns: non-significant. (**E**) Relative expression levels of immune cell markers (*Tbx21/β-actin, Gata3/β-actin, Rorc/β-actin, Foxp3/β-actin, Cebpe/β-actin*) for analyzing immune cell populations measured by RT-qPCR in spleen. n≥5. Data are represented as mean ± SEM; *p<0.05; **p<0.01; ***p<0.001; ns: non-significant. (**F**) H&E staining of spleen sections. Scale bar: 500 µm.

The online version of this article includes the following source data and figure supplement(s) for figure 7:

**Source data 1.** Excel file containing qPCR data for *Figure 7A, C, D and E*.

**Source data 2.** Original files of Western blot analysis for *Figure 7B*.

**Source data 3.** PDF file containing Western blot analysis for *Figure 7B*, indicating relevant bands.

**Figure supplement 1.** Cytokine quantification in plasma measured by the Mouse Cytokine/Chemokine 44-Plex Discovery Assay.

**Figure supplement 1—source data 1.** Excel file containing Cytokine quantification data for *Figure 7—figure supplement 1*.

**Figure supplement 2.** Differential expression analysis of quantitative proteomic data from the spleen.

normalized the levels of IL-5 and CCL2 in plasma (*Figure 7C*) and reduced the levels of *IL-5* transcripts in spleen (*Figure 7D*). The levels of a range of other cytokines in plasma did not change (*Figure 7—figure supplement 1*). We determined the proportion of immune cell populations in the spleen by using gene expression markers and found a significant reduction of all T helper cell lineage markers (*T-Box Transcription Factor 21* (*Tbx21*), *GATA Binding Protein 3* (*Gata3*), *RAR Related Orphan Receptor C* (*Rorc*), and *Forkhead Box P3* (*Foxp3*)) in mtDNA mutator spleen. In contrast, a granulocyte marker (*CCAAT Enhancer Binding Protein Epsilon, Cebpe*) was increased (*Figure 7E*). This finding implies a significant shift in immune cell populations in the spleen of *Polg*$^{-/mut}$; *Tfam*$^{+/+}$ mice. TFAM overexpression did not affect the expression of the immune cell markers in the spleen. The mtDNA mutator mice have been shown to have disrupted white pulp structure in the spleen due to persistent inflammation (*Lei et al., 2021*). However, the overall histology of the spleen did not reveal any apparent rescue effect by TFAM modulation (*Figure 7F*).

Finally, we subjected spleen samples to tandem mass tag-based quantitative proteomic analysis to reveal any further beneficial or detrimental effects of TFAM modulation. In comparison with wild-type mice, the *Polg*$^{-/mut}$; *Tfam*$^{+/+}$ spleen showed upregulation of proteins involved in heme biosynthesis and reactive oxygen species (ROS) defense pathways. The former is consistent with the severe anemia present in the mtDNA mutator mice (*Balducci, 2003*), causing compensatory extramedullary haematopoiesis (*Trifunovic et al., 2004*). The latter is possibly due to altered ROS signaling, which is an important player in reshuffling hematopoietic cell populations (*Ahlqvist et al., 2015*). Increasing TFAM levels mitigated the induction of both of these pathways (*Figure 7—figure supplement 2*). The beneficial effect can potentially be attributed to an acute improvement of OXPHOS function due to the increased levels of wild-type mtDNA segments, to a long-term proliferative advantage and hence clonal expansion of cells harboring less mutated mtDNA molecules, or a combination of both.

To summarize, TFAM elevation evokes corresponding changes in mtDNA levels in the spleen, while mtDNA gene expression is unaffected. The absolute increase of mtDNA seen in TFAM overexpressing mice leads to a positive effect on spleen physiology, in line with previous findings in the testis of mtDNA mutator mice (*Jiang et al., 2017*).

## Discussion

We demonstrate here that even moderate changes in TFAM levels can have drastically different outcomes on the premature ageing phenotypes of mtDNA mutator mice. Depending on the tissue, an increase in TFAM levels can be detrimental or beneficial. Likewise, reducing TFAM levels also shows ambivalent effects. Whereas most tissues did not seem to be affected by reduced TFAM levels, BAT gene expression was reconstituted, indicating normalized physiology. This highlights the complexity and tissue specificity of the regulation of mtDNA copy number and expression and warrants careful evaluation of the therapeutic potential of TFAM alterations in different disease settings.

Somatic mutations of mtDNA are important contributors to the ageing process and are found in various common age-associated diseases, including cancer, neurodegeneration, diabetes, and cardiac disorders (*Wallace, 2010*). They act by causing focal respiratory chain dysfunction, eventually

compromising cell function and tissue homeostasis (*Bender et al., 2006*; *Müller-Höcker, 1989*; *Müller-Höcker, 1990*; *Taylor et al., 2003*). Several elegant strategies have been developed to ameliorate the accompanying pathology. One strategy has focused on increasing mitochondrial mass by overexpressing PGC-1α and has been effective in improving the skeletal muscle and heart phenotypes of the mtDNA mutator mouse (*Dillon et al., 2012*). However, PGC-1α levels must be carefully titrated as it is involved in several other cellular processes, and aberrant activation of PGC-1α can be harmful (*Miura et al., 2006*; *Ciron et al., 2012*; *Filograna et al., 2021*). Similarly, tampering with mitochondrial mass might be problematic given the diverse functional roles of mitochondria. Another promising and more direct strategy is to increase the mtDNA copy number, and thereby the absolute amount of wild-type mtDNA molecules, by raising TFAM levels. Notably, TFAM modulation is not without risk because mtDNA copy number does follow TFAM levels only up until a certain point, which seems to be tissue-specific. Increasing TFAM levels beyond this point does not result in a further rise in mtDNA copy number, but instead increases the TFAM-to-mtDNA ratio (*Bonekamp et al., 2021*). The TFAM-to-mtDNA ratio determines nucleoid compaction, which, in turn, is crucial for mtDNA gene expression. Strong TFAM overexpression can, therefore, lead to impaired mtDNA expression and respiratory chain deficiency in a tissue-specific manner, and eventually shorten the life span of the mouse (*Bonekamp et al., 2021*). Nonetheless, moderate TFAM overexpression by about 50% has been shown to be well tolerated in all mouse models investigated so far (*Jiang et al., 2017*; *Bonekamp et al., 2021*; *Filograna et al., 2019*). Indeed, this rationale also demonstrated to be effective in rescuing the early-onset male infertility phenotype in mtDNA mutator mice of 4 months of age (*Jiang et al., 2017*). The TFAM increase resulted in doubling of the mtDNA copy number in mouse testis. While the total mtDNA mutation load was unaltered, the increased mtDNA levels thus led to a higher absolute number of mtDNA molecules carrying segments without mutations. Likewise, increasing the absolute number of wild-type mtDNA copies via moderately increasing TFAM expression in the m.C5024T tRNA^Ala mouse model showed a beneficial effect on the cardiomyopathy phenotype in aged mice (*Filograna et al., 2019*).

We did not observe ubiquitously beneficial effects when employing moderate TFAM overexpression to alleviate the premature ageing phenotypes of mtDNA mutator mice. On the one hand, TFAM elevation in the spleen led to a concomitant increase in full-length mtDNA translating into improved spleen homoeostasis. On the other hand, increasing TFAM levels in liver did not result in higher mtDNA levels but instead caused impaired mtDNA expression. The tissue-specific responses in mtDNA mutator mice differ from what was seen in mice without mtDNA mutations or in mice carrying the heteroplasmic pathogenic m.C5024T tRNA^Ala mutation as well as mice carrying an mtDNA deletion (*Nishiyama et al., 2010*). The response in mtDNA mutator mice resembles the tissue-specific effects seen in mice with very strong TFAM overexpression, where the mtDNA levels follow TFAM levels only up to a certain point in a tissue-specific manner. The discrepancy in outcomes of moderate TFAM overexpression can potentially be explained by the varying presence of compensatory mechanisms attributed to different levels of OXPHOS malfunction in the investigated mouse models. The m.C5024T tRNA^Ala mouse model harbors a heteroplasmic maternally inherited pathogenic mtDNA mutation that at high levels causes a mitochondrial translational defect which mildly affects OXPHOS protein levels and function (*Filograna et al., 2019*; *Kauppila et al., 2016*). In contrast, the mtDNA mutator mouse continuously generates new mtDNA mutations every time an mtDNA molecule is replicated and it is, therefore, much more severely affected. The abundant mtDNA mutations in the mtDNA mutator mice affect the tRNAs and rRNAs needed for mitochondrial translation and cause abundant amino acid substitutions in the mtDNA-encoded OXPHOS subunits. The OXPHOS deficiency in the mtDNA mutator mice is thus caused by a combination of reduced synthesis of mtDNA-encoded OXPHOS subunits and synthesis of mtDNA-encoded OXPHOS subunits with defective function, which will reduce both the levels and the function of the OXPHOS complexes. The more pronounced pathology in mtDNA mutator mice likely explains the activation of compensatory mechanisms consistent with the intrinsic upregulation of TFAM found in the heart. As such, TFAM is necessary but not sufficient to control mtDNA levels under pathological conditions on its own, but other licensing factors seem to be limiting. While TFAM seems to be the main protein coating mtDNA, the precise stoichiometry and positioning likely vary between different cellular states, providing access for other proteins such as the mitochondrial single-stranded DNA binding protein (mtSSB) (*Brüser et al., 2021*; *Wang et al.,*

*2013*; *Blumberg et al., 2018*; *Isaac et al., 2024*; *Jiang et al., 2021*). Additionally, further licensing factors could, for example, include the abundance of other proteins involved in mtDNA replication and expression and go in line with previous studies in cultured mammalian cells as well as budding yeast (*Maniura-Weber et al., 2004*; *Seel et al., 2023*; *Kozhukhar and Alexeyev, 2019*). Notably, the severity of the OXPHOS dysfunction largely differed between tissues, and accordingly, the endogenous compensatory mechanisms as well as the rescuing efficacy of TFAM modulation was tissue-specific. The different vulnerability of a given tissue towards OXPHOS dysfunction caused by mtDNA mutations is likely dependent on its physiology, metabolism, proliferative character, absolute mtDNA levels, mtDNA turnover rates, the TFAM-to-mtDNA ratios, and additional factors.

In summary, the mtDNA mutator mouse shows tissue-specific endogenous compensatory mechanisms in response to the continuous mutagenesis of mtDNA, including the upregulation of TFAM expression, elevated mtDNA copy number, as well as altered mtDNA gene expression. This likely limits the impact of genetic manipulation of TFAM expression and explains the variable tissue-specific consequences of altered TFAM expression on mtDNA copy number, mtDNA gene expression, and tissue physiology. Importantly, this also clearly demonstrates that TFAM is not the only determinant of mtDNA levels under pathological conditions, but argues that other factors involved in mtDNA replication must play a limiting role. Identifying these additional molecular players holds the promise to more accurately intervene with the ageing process and counteract other pathological entities caused by mtDNA mutations, including mitochondrial disorders, cancer, neurodegenerative diseases, diabetes, and cardiac disorders.

# Materials and methods

## Key resources table

| Reagent type (species) or resource | Designation | Source or reference | Identifiers | Additional information |
|---|---|---|---|---|
| Genetic reagent (*M. musculus*) | Mouse: *Polg*$^{+/mut}$ | *Trifunovic et al., 2004* | NA | |
| Genetic reagent (*M. musculus*) | Mouse: *Polg*$^{+/-}$ | *Hance et al., 2005* | NA | |
| Genetic reagent (*M. musculus*) | Mouse: *Tfam*$^{+/OE}$ | *Jiang et al., 2017* | NA | |
| Genetic reagent (*M. musculus*) | Mouse: *Tfam*$^{+/-}$ | *Larsson et al., 1998* | NA | |
| Antibody | Anti-HSP60 antibody | Enzo Lifesciences | AB1-SPA-807-E | WB |
| Antibody | Total OXPHOS Rodent WB Antibody Cocktail | Abcam | ab110413 | WB |
| Antibody | Anti-TFAM antibody | Abcam | ab131607 | WB |
| Antibody | Anti-UCP1 antibody | Abcam | ab209483 | WB |
| Antibody | Anti-COX2 antibody | home made | This study | WB |
| Commercial assay or kit | Fibroblast Growth Factor 21 Mouse/Rat ELISA | BioVendor | UNQ3115/PRO10196 | |
| Commercial assay or kit | Mouse Cytokine/Chemokine 44-Plex Discovery Assay Array | Eve Technologies | MD44 | |
| Software, algorithm | Prism 9 | Graph Pad | https://www.graphpad.com/ | |
| Software, algorithm | Image J | NIH | https://imagej.net/ij/ | |
| Software, algorithm | Adobe Photoshop 2020 | Adobe | https://www.adobe.com/home | |
| Software, algorithm | Adobe Illustrator 2020 | Adobe | https://www.adobe.com/home | |

*Continued on next page*

*Continued*

| Reagent type (species) or resource | Designation | Source or reference | Identifiers | Additional information |
|---|---|---|---|---|
| Software, algorithm | Proteome Discoverer v2.4 | Thermo Fisher Scientific | https://www.thermofisher.com/de/de/home/industrial/mass-spectrometry/liquid-chromatography-mass-spectrometry-lc-ms/lc-ms-software/multi-omics-data-analysis/proteome-discoverer-software.html | |
| Software, algorithm | R (4.1.0) | R project | https://www.r-project.org/ | |
| Other | Mass spectrometry proteomics data | This paper | ProteomeXchange Consortium (PRIDE partner repository): PXD054598 | |

## Mouse work

The heterozygous *Polg*$^{+/mut}$ mutator mice, the heterozygous *Polg*$^{+/-}$, the *TFAM BAC* (*Tfam*$^{+/OE}$) and *Tfam*$^{+/-}$ (Jax MGI ID: 1860962) mice were generated previously as described (*Trifunovic et al., 2004*; *Jiang et al., 2017*; *Larsson et al., 1998*; *Hance et al., 2005*). Heterozygous *Polg* knockout (*Polg*$^{+/-}$) males were mated to females that carry either the *Tfam*$^{+/OE}$ or *Tfam*$^{+/-}$ allele, the resulting *Polg*$^{+/-}$; *Tfam*$^{+/OE}$ or *Polg*$^{+/-}$; *Tfam*$^{+/-}$ females were further mated to heterozygous mtDNA mutator (*Polg*$^{+/mut}$) males to generate the four genotypes used in this study (illustrated in *Figure 1—figure supplement 1*). Transgenic mice on a pure C57BL/6 N background were housed in a 12 hr light/dark cycle in standard individually ventilated cages and fed ad libitum with a normal chow diet. Experimental groups included only 35-weeks-old male animals. The study was approved by the Stockholm animal welfare ethics committee (Stockholms djurförsöksetiska nämnd) under the ethical permit 2001–2018 and carried out following the guidelines of the Federation of European Laboratory Animal Science Associations (FELASA).

## Tissue isolation

Animals were euthanized by $CO_2$ followed by cervical dislocation. Blood was taken by heart puncture, collected in EDTA tubes, and centrifuged at 2000 *g* and 4 °C for 10 min to separate the plasma. Heart, spleen, liver, testis, colon, and BAT were collected immediately, washed with phosphate-buffered saline (PBS), and a piece of each tissue snap-frozen in liquid nitrogen, and stored at –80 °C. For hematoxylin and eosin (H&E) staining, tissues were embedded in O.C.T. compound (Tissue-Tek), frozen in isopentane precooled in liquid nitrogen, and stored at –80 °C. For the liver and colon, an additional tissue piece was kept in PBS for subsequent isolation of crude mitochondria.

## Mitochondrial isolation

Liver and colon were homogenized in mitochondrial isolation buffer containing 320 mM sucrose, 1 mM EDTA, and 10 mM Tris-HCl, pH 7.4, supplemented with 0.2% bovine serum albumin (Sigma-Aldrich), EDTA-free complete protease inhibitor cocktail, and PhosSTOP tablets (Roche) by using a Teflon pestle (Schuett Biotec). After centrifugation at 1000 × g (swing-out rotor) for 10 min at 4 °C, the supernatants were subsequently spun at 10,000 × g for 10 min at 4 °C to isolate the mitochondria. Crude mitochondrial pellets were resuspended in a suitable amount of mitochondrial isolation buffer.

## DNA isolation and mtDNA quantification by qPCR

Genomic DNA from snap-frozen heart, spleen, liver, testis, colon, and BAT was isolated using the DNeasy Blood and Tissue Kit (Qiagen), following the manufacturer's instructions. Quantification of mtDNA copy number was performed in triplicates using 5 ng of DNA using TaqMan Universal Master Mix II and TaqMan probes (Life Technologies). The mtDNA levels were assessed using probes against

the mitochondrial genes encoding *Nd1*, *Atp6*, and *Cytb* and normalized to the nuclear gene encoding *18* S rDNA.

## DNA isolation and mtDNA quantification by Southern blot analysis

Genomic DNA from snap-frozen liver was isolated using the Puregene Cell and Tissue Kit (Qiagen), following the manufacturer's instructions. Southern blot analysis was performed as described previously using CytB to detect mtDNA and 18 S rDNA as nuclear loading control (*Jiang et al., 2019*).

For mouse samples, 2 µg total genomic DNA was digested with SacI-HF at 37 °C overnight and preheated at 93 °C for 3 min, followed by cooling on ice before loading onto the gel. After electrophoresis in 0.8% agarose, DNA was depurinated by incubation in 0.25 M HCl for 10 min and incubated in denaturation buffer (0.5 M NaOH and 1.5 M NaCl) twice for 30 min and neutralization buffer (0.5 M Tris-HCl (pH 7.4) and 1.5 M NaCl) twice for 30 min. DNA was blotted onto a Hybond N+nitrocellulose membrane for 72 hr and then cross-linked by exposure to 254 nm ultraviolet, 200 mJ/cm$^2$. Next, membranes were hybridized with α-[$^{32}$P]-dCTP-labelled DNA probes to detect mtDNA (CytB) or nuclear 18 S rDNA as a loading control. Radioactive signals were visualized using PhosphorImager screens and a Typhoon 7000 FLA (GE Healthcare). Band intensities were quantified using ImageJ software.

## RNA isolation and quantitative reverse transcription PCR

Total RNA from snap-frozen heart, spleen, liver, colon, and BAT was isolated using the TRIzol/chloroform extraction method and quantified with a Qubit fluorometer (Life Technologies). After deoxyribonuclease treatment, reverse transcription was performed using the High-Capacity cDNA Reverse Transcription Kit (Applied Biosystems, Life Technologies). RT-qPCR was performed using the TaqMan Universal Master Mix II with TaqMan probes for *Nd1*, *Atp6*, *Cytb*, *Nppa*, *Atf4*, *Atf5*, *Mthfd2*, *Ucp1*, *Il-5*, *Ccl2,* and *β-actin* (Life Technologies). β-*actin* was used as the loading control.

## Western blot

Tissues were homogenized in RIPA buffer (50 mM Tris pH 7.4, 150 mM NaCl, 1% Nonidet P-40, 0.5% DOC, 0.1% SDS) on ice. The supernatant was collected after centrifugation at 10,000 g for 10 min at 4 °C. Protein concentration was determined by the BCA assay. After mixing with NuPAGE LDS Sample Buffer (Invitrogen), 20 µg of total tissue lysate was loaded into 12% precast gels (Invitrogen) and separated by SDS-PAGE. Protein was transferred onto polyvinylidene difluoride membranes using the iBlot 2 Gel Transfer system (Invitrogen). Immunodetection was performed according to standard procedure using enhanced chemiluminescence (Clarity ECL Western Blotting Substrates, Bio-Rad) and imaged using the Bio-Rad ChemDoc system. Images were exported using the Bio-Rad ImageLab software and quantified using Image J. The following antibodies were used: HSP60 (Enzo Lifesciences AB1-SPA-807-E), Total OXPHOS Rodent WB Antibody Cocktail (ab110413, abcam), TFAM (ab131607, abcam), UCP1 (ab209483, abcam), and COX2 (rabbit polyclonal antisera against COX2 generated using recombinant mouse protein).

## TFAM-to-mtDNA ratios calculation

Average TFAM levels were quantified using Image J and normalized to HSP60 levels (n=2). To compare between the different groups, the averaged data was normalized to the wild-type (*Polg$^{+/+}$Tfam$^{+/+}$*). For the mtDNA level, qPCR data from the ND1 probe was used (n ≥ 5). For group comparisons, the averaged data was normalized to the wild-type (*Polg$^{+/+}$Tfam$^{+/+}$*). Eventually, TFAM-to-mtDNA ratios were calculated by dividing the wild-type normalized TFAM levels by the wild-type normalized mtDNA levels.

## Histochemistry

For H&E staining, the spleen was cryosectioned at −20 °C (10 µm section; Cryostar NX70-Thermo Fisher) onto Polysine-coated slides (VWR 631–0107) and stored at −80 °C until use. H&E staining was performed according to standard procedure. In short, slides were brought to room temperature, washed once in water, and stained with hematoxylin solution (ab220365, abcam) for 10 min. The slides were washed in water twice and once in 0.1% sodium bicarbonate. After washing in 96% ethanol, the

slides were stained in Eosin Y solution (0.25%) for 5 s. The slides were washed in ethanol, dehydrated, and mounted for bright-field microscopy.

## OXPHOS activity measurements

Spectrophotometric assessment of OXPHOS enzyme activities was performed as previously described (*Wibom et al., 2002*). In brief, 500 µg isolated mitochondria were resuspended in 100 µL resuspension buffer (250 mM sucrose, 15mM $KH_2PO_4$, 2 mM MgAc$_2$, 0.5 mM EDTA, 0.5 g/l HSA, pH 7.2) and stored as 10 µL aliquots at −80 °C. All assays were performed using an Indiko automated photometer (Thermo Fisher Scientific) fitted with filters for 340, 405, 550, and 600 nm (bandwidth±5 nm) at 37 °C.

For activity measurements of NADH:coenzyme Q reductase (complex I) and NADH:cytochrome c reductase (complex I + III), samples were pretreated by adding 590 µL of a solution of 5 mM $KH_2PO_4$, 5 mM $MgCl_2$, 0.5 g/l HAS (pH 7.2) to the 10 µL frozen mitochondrial suspension. Within 1 min, another 50 µL of the same solution supplemented with 7.15 g/L saponin was added.

For measurement of complex I activity, 72 µL pretreated mitochondria were incubated for 7 min in a reaction mixture with a final composition of 50 mM $KH_2PO_4$, 5 mM $MgCl_2$, 5 g/L HSA, 0.2 mM KCN, 1.2 mg/L antimycin A, and 0.12 mM coenzyme $Q_1$ (pH 7.5). Subsequently, NADH was added to a final concentration of 0.15 mM, and the decrease in absorbance was monitored for 1 min before and after the addition of 2 mg/L rotenone at 340 nm in a final volume of 150 µL. For the rotenone-sensitive activity calculation, an extinction coefficient of 6.81 L/mmol/cm was used. For measurement of complex I + III activity, 6 µL pretreated mitochondria were incubated for 7 min in a reaction mixture with the final composition of 50 mM $KH_2PO_4$, 5 mM $MgCl_2$, 5 g/L HAS, 0.2 mM KCN, and 0.12 mM cytochrome c (oxidized form) (pH 7.5). Subsequently, NADH was added to a final concentration of 0.15 mM, and the increase in absorbance was monitored for 1 min before and after the addition of rotenone, 2 mg/L at 550 nm in a final volume of 125 µL.

For activity measurements of succinate dehydrogenase (complex II) and succinate:cytochrome c reductase (complex II + III), samples were pretreated by incubating 10 µL of the frozen mitochondrial suspension for 30 min in 100 µL of 50 mM $KH_2PO_4$, 30 mM succinate, 7.5 mM $MgCl_2$, and 0.45 g/L saponin (pH 7.2) at 37°C. Complex II activity was determined as previously described (*Birch-Machin et al., 1994*). 10 µL pretreated mitochondria were incubated for 15 min in a reaction mixture of 20 mM $KH_2PO_4$, 5 mM $MgCl_2$, 25 mM succinate, 0.2 mM KCN, 0.05 mM 2,6-dichloroindophenol (DCIP), and 2 mg/L antimycin A (pH 7.5). Subsequently, the blank rate was measured for 1 min followed by Coenzyme $Q_1$ addition to a final concentration of 0.05 mM and monitoring the decrease in absorbance at 600 nm for 1 min in a final volume of 150 µL. For activity calculation, an extinction coefficient of 22 L/mmol/cm was used.

Measurement of complex II + III activity was performed by measuring the blank rate in 50 mM $KH_2PO_4$, 5 mM $MgCl_2$, 5 g/L HSA, 0.2 mM KCN, 30 mM succinate, 2 mg/L rotenone, and 0.12 mM cytochrome c (oxidized form) (pH 7.5). Subsequently, 5 µL pretreated mitochondria were added and the reduction of cytochrome c was monitored for 2 min at 550 nm in a final volume of 150 µL.

For cytochrome c oxidase (complex IV) activity measurement, pretreatment was performed by diluting mitochondria to a concentration corresponding to 100U/L of citrate synthase in 1 g/L digitonin and 50 mM $KH_2PO_4$ (pH 7.5). The blank rate was recorded in a solution of 50 mM $KH_2PO_4$, 2 mg/L rotenone, and 0.03 mM cytochrome c (reduced form) (pH 7.5). Subsequently, 10 µL of pretreated mitochondria were added and oxidation of cytochrome c was followed for 1 min at 550 nm in a final volume of 250 µL.

Citrate synthase activity was used as a mitochondrial marker and determined as previously described (*Alp et al., 1976*). 10 µL frozen mitochondria were pretreated by adding 240 µl of 50 mM $KH_2PO_4$, 1 mM EDTA, 0.1% Triton X-100 (pH 7.5). 20 µL pretreated mitochondria were then incubated in 50 mM Tris, 0.20 mM 5,5′-Dithiobis(2 nitrobenzoic acid) (DTNB), 0.1 mM Acetyl-CoA (pH 8.1) for 5 min. Subsequently, oxaloacetic acid was added to a final concentration of 0.5 mM and the increase in absorbance was monitored at 405 nm for 1 min in a final volume of 250 µL. For activity calculation, an extinction coefficient of 13.6 L/mmol/cm was used.

For BN-PAGE, 100 µg of isolated mitochondria were lysed in 50 µL NativePAGE Sample Buffer (Invitrogen) containing 1% (w/v) digitonin (Calbiochem). Samples were incubated for 10 min at 4 °C and centrifuged at 17,000 g for 30 min. After centrifugation, supernatants were collected and protein concentration was measured using the BCA assay. 1/10 volume of 5% Coomassie blue G-250

(Invitrogen BN2004) was added to the supernatant. 15 µg protein was loaded on 3–12% Bis-Tris NativePAGE gels (Invitrogen) and run according to the manufacturer's instructions.

For CI in gel activity measurements, the BN-PAGE gel was incubated in 2 mM Tris/HCl pH 7.4, 0.1 mg/mL NADH (Roche), and 2.5 mg/mL nitrotetrazolium blue for blue staining (Sigma) for about 10 min. CIV in gel activity was determined by incubating the BN-PAGE gels in 10 mL of 0.05 mM phosphate buffer pH 7.4, 25 mg 3.3'-diamidobenzidine tetrahydrochloride (DAB), 50 mg Cyt C, 3.75 g Sucrose, and 1 mg Catalase for approximately 1 hr.

## Immune profiling

For cytokine measurements, plasma was sent to Eve Technologies for assessment (Mouse Cytokine/Chemokine 44-Plex Discovery Assay Array (MD44)).

For FGF21 measurement, the ELISA kit from Biovendor (UNQ3115/PRO10196) was used following the manufacturer's protocol.

## Quantitative mass spectrometry

Mouse spleen tissue was prepared as described previously (*Clemente et al., 2022*) with some modifications. In brief, samples were thawed on ice and 20–25 mg tissue was cut into small pieces and supplemented with 50 µL of 8 M urea, 50 µL of 0.2% ProteaseMAX (Promega) in 20% acetonitrile (ACN), 100 mM Tris-HCl, pH 8.5 and 100 mM NaCl and 1 µL of 100 x protease inhibitor (Pierce) before transferring to a prefilled tube containing 400 µm LoBind silica beads. The samples were frozen for a short time before homogenization using a Disruptor Genie at maximal speed on 2800 rpm for 2 min, incubated on ice for 2 min. These steps were repeated five times. The samples were then centrifuged at 13,000 g for 10 min at 4 °C. The supernatant was collected and 100 µL of Tris-HCl was used to wash the beads, which was combined with the supernatant. Proteins were precipitated with fourfold volumes of chilled acetone before protein concentration was determined by BCA assay (Pierce). An aliquot of 30 µg samples was reduced with 2.5 µL of 250 mM dithiothreitol, alkylated with 3 µL of 500 mM chloroacetic acid, and digested by addition of 0.6 µg of sequencing grade modified trypsin (Promega) and incubation at 37 °C for 16 hr. The digestion was stopped with 4.5 µL cc of formic acid and incubating the solution at room temperature (RT) for 5 min. The sample was cleaned on a C18 Hypersep plate with 40 µL bed volume (Thermo Fisher Scientific) and dried using a vacuum concentrator (Eppendorf). Biological samples were labeled with TMTpro reagents (Thermo Fisher Scientific) in random order, adding 100 µg TMT-reagent in 30 µL anhydrous ACN to each digested sample resolubilized in 85 µL of 50 mM triethylammonium bicarbonate and incubated at RT for 2 h. The labeling reaction was stopped by adding 11 µL of 5% hydroxylamine and incubating at RT for 15 min before combining all 15 biological samples in one vial. The sample was cleaned on a C18 Hypersep plate with 40 µL bed volume (Thermo Fisher Scientific) and dried using a vacuum concentrator (Eppendorf). For fractionation, the TMT-labeled peptides were dissolved in 50 µL of 20 mM ammonium hydroxide and were loaded onto an XBridge bridged ethyl hybrid C18 UPLC column (2.1 mm inner diameter ×250 mm, 2.5 µm particle size, Waters), and profiled with a linear gradient of 5–60% 20 mM ammonium hydroxide in ACN (pH 10.0) over 48 min, at a flow rate of 200 µL/min. The chromatographic performance was monitored by sampling eluate with a UV detector (UltiMate 3000 UPLC, Thermo Fisher Scientific) scanning at 214 nm. Fractions were collected at 30 s intervals into a 96-well plate and combined in 12 samples concatenating 8–8 fractions representing peak peptide elution before drying in a vacuum concentrator. Peptides were reconstituted in solvent A (2% ACN, 0.1% FA) and approx. 2 µg samples injected on a 50 cm long EASY-Spray C18 column (Thermo Fisher Scientific) connected to an UltiMate 3000 nanoUPLC system (Thermo Fisher Scientific) using a 90 min long gradient: 4–26% of solvent B (98% ACN, 0.1% FA) in 90 min, 26–95% in 5 min, and 95% of solvent B for 5 min at a flow rate of 300 nL/min. Mass spectra were acquired on a Q Exactive HF hybrid quadrupole Orbitrap mass spectrometer (Thermo Fisher Scientific) ranging from $m/z$ 375–1500 at a resolution of $R$=120,000 (at $m/z$ 200) targeting $5\times10^6$ ions for maximum injection time of 80ms, followed by data-dependent higher-energy collisional dissociation (HCD) fragmentations of precursor ions with a charge state of 2+ to 7+, using 45 s dynamic exclusion. The tandem mass spectra of the top 18 precursor ions were acquired with a resolution of $R$=60,000,, targeting $2\times10^5$ ions for a maximum injection time of 54 ms, setting quadrupole isolation width to 1.4 Th and normalized collision energy to 33%. Acquired raw data files were analyzed using Proteome Discoverer v2.4 (Thermo Fisher Scientific) with the Mascot

Server v2.5.1 (Matrix Science Ltd., UK) search engine against mouse protein database (SwissProt). A maximum of two missed cleavage sites was allowed for full tryptic digestion, while setting the precursor and the fragment ion mass tolerance to 10 ppm and 0.02 Da, respectively. Carbamidomethylation of cysteine was specified as a fixed modification, while TMTpro on lysine and N-termini, oxidation on methionine as well as deamidation of asparagine and glutamine were set as dynamic modifications. Initial search results were filtered with 5% FDR using the Percolator node in Proteome Discoverer. Quantification was based on the TMT-reporter ion intensities. The data was analyzed using R (4.1.0) using the preprocessCore (*Bolstad, 2024*) and pheatmap packages.

## Quantification and statistical analysis

The sample size is indicated in the figure legends and included at least five mice where statistical evaluation was performed. ImageJ was used for quantification. Statistical analysis and generation of graphs were performed with GraphPad Prism v9 software, except for quantitative mass spectrometry data, which was analyzed and plotted using R as described above. Statistical comparisons were performed using one-way analysis of variance (ANOVA), and post hoc analysis was conducted with Dunnett's multiple comparisons test. Values of $p < 0.05$ were considered statistically significant. Images were processed with Adobe Photoshop 2020 and schematics were created with Adobe Illustrator 2020.

## Acknowledgements

LSK was supported by an EMBO long-term fellowship (ALTF 570–2019). NGL was supported by the Swedish Research Council (2015–00418), the Swedish Cancer Foundation (21 1409 Pj), the Knut and Alice Wallenberg Foundation (2019.0109 and 2023.0224), the Swedish Brain Foundation (FO2021-0080), the Swedish Diabetes Foundation (DIA2020-516 and DIA2021-620), the Novo Nordisk Foundation (NNF20OC006316, NNF22OC0078444), and by grants provided by Region of Stockholm (ALF project).

## Additional information

### Competing interests

Nils-Göran Larsson: is the inventor of the C5024T mutant mouse licensed to the pharmaceutical industry by the Max Planck Society, and is a scientific founder of Pretzel Therapeutics Inc and owns stock in this company. The other authors declare that no competing interests exist.

### Funding

| Funder | Grant reference number | Author |
| --- | --- | --- |
| European Molecular Biology Organization | ALTF 570-2019 | Laura Sophie Kremer |
| Vetenskapsrådet | 2015-00418 | Nils-Göran Larsson |
| Cancerfonden | 21 1409 Pj | Nils-Göran Larsson |
| Knut och Alice Wallenbergs Stiftelse | 2019.0109 | Nils-Göran Larsson |
| Hjärnfonden | FO2021-0080 | Nils-Göran Larsson |
| Diabetesfonden | DIA2020-516 | Nils-Göran Larsson |
| Novo Nordisk Foundation Center for Basic Metabolic Research | NNF20OC006316 | Nils-Göran Larsson |
| Knut och Alice Wallenbergs Stiftelse | 2023.0224 | Nils-Göran Larsson |
| Diabetesfonden | DIA2021-620 | Nils-Göran Larsson |

| Funder | Grant reference number | Author |
|---|---|---|
| Novo Nordisk Foundation Center for Basic Metabolic Research | NNF22OC0078444 | Nils-Göran Larsson |

The funders had no role in study design, data collection and interpretation, or the decision to submit the work for publication.

## Author contributions

Laura Sophie Kremer, Conceptualization, Formal analysis, Supervision, Funding acquisition, Visualization, Methodology, Writing – original draft, Writing – review and editing; Guanbin Gao, Formal analysis, Visualization, Methodology, Writing – original draft; Giovanni Rigoni, Roberta Filograna, Mara Mennuni, Rolf Wibom, Ákos Végvári, Formal analysis; Camilla Koolmeister, Conceptualization, Supervision; Nils-Göran Larsson, Conceptualization, Supervision, Funding acquisition, Writing – original draft, Writing – review and editing

## Author ORCIDs

Laura Sophie Kremer https://orcid.org/0000-0002-1647-7538
Mara Mennuni https://orcid.org/0000-0001-6199-6233
Rolf Wibom https://orcid.org/0000-0001-6721-4642
Nils-Göran Larsson https://orcid.org/0000-0001-5100-996X

## Ethics

The study was approved by the Stockholm animal welfare ethics committee (Stock-holms djurförsöksetiska nämnd) under the ethical permit #2001-2018 and carried out following the guidelines of the Federation of European Laboratory Animal Science Associations (FELASA).

Reviewer #1 (Public review): https://doi.org/10.7554/eLife.104461.3.sa1
Reviewer #2 (Public review): https://doi.org/10.7554/eLife.104461.3.sa2
Author response https://doi.org/10.7554/eLife.104461.3.sa3

# Additional files

## Supplementary files

MDAR checklist

## Data availability

The mass spectrometry proteomics data have been deposited to the ProteomeXchange Consortium via the PRIDE partner repository (*Perez-Riverol et al., 2022*) with the dataset identifier PXD054598. Proteomics data was analyzed using standard packages and did not generate any new code. Any additional information required to reanalyze the data reported in this paper is available from the lead contact upon request.

The following dataset was generated:

| Author(s) | Year | Dataset title | Dataset URL | Database and Identifier |
|---|---|---|---|---|
| Kremer LS, Gao G, Rigoni G, Filograna R, Mennuni M, Wibom R, Vegvari A, Koolmeister C, Larsson NG | 2025 | Tissue-specific responses to TFAM and mtDNA copy number manipulation in prematurely ageing mice | https://proteomecentral.proteomexchange.org/cgi/GetDataset?ID=PXD054598 | ProteomeXchange, PXD054598 |

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
