## [Editor Report · eLife Assessment]

This is an **important** study that examines the role of TFAM, a protein that helps maintain mtDNA, in mtDNA mutator mice. With **convincing** evidence, the authors have demonstrated that TFAM's counteractive role in mtDNA mutator mice is tissue-specific. The study does a thorough job of assessing the impact of modulating TFAM levels in a polg mutator mouse model of aging. The authors have thoroughly addressed all the points raised during the first round of review.

---

## [Referee Report · Reviewer #1 (Public review)]

Summary:

This manuscript by Kremer et al. characterizes the tissue-specific responses to changes in TFAM levels and mtDNA copy number in prematurely aging mice (polg mutator model). The authors find that overexpression of TFAM can have beneficial or detrimental effects depending on the tissue type. For instance, increased TFAM levels increase mtDNA copy number in the spleen and improve spleen homeostasis but do not elevate mtDNA copy number in the liver and impair mtDNA expression. Similarly, the consequences of reduced TFAM expression are tissue-specific. Reduced TFAM levels improve brown adipocyte tissue function while other tissues are unaffected. The authors conclude that these tissue-specific responses to altered TFAM levels demonstrate that there are tissue-specific endogenous compensatory mechanisms in response to the continuous mutagenesis produced in the prematurely aging mice model, including upregulation of TFAM expression, elevated mtDNA copy number, and altered mtDNA gene expression. Thus, the impact of genetically manipulating global TFAM expression is limited and there must be other determinants of mtDNA copy number under pathological conditions beyond TFAM.

Strengths:

Overall, this is an interesting study. It does a good job of demonstrating that given the multi-functional role of TFAM, the outcome of manipulating its activity is complex.

Weaknesses:

No major weaknesses noted. The authors have adopted all our suggestions to improve the clarity of the manuscript.

---

## [Referee Report · Reviewer #2 (Public review)]

Summary:

This study by Kremer et al. investigates the impact of modulation of expression of TFAM, a key protein involved in mitochondrial DNA (mtDNA) packaging and expression, in mtDNA mutator mice, which carry random mtDNA mutations. While previous research suggested that increasing TFAM could counteract the pathological effects of mtDNA mutations, this study reveals that the effects of TFAM modulation are tissue-specific. These findings highlight the complexity of mtDNA copy number regulation and gene expression, emphasizing that TFAM alone is not the sole determinant of mtDNA levels in contexts where oxidative phosphorylation is impaired. Other factors likely play a significant role, underscoring the need for nuanced approaches when targeting TFAM for therapeutic interventions.

Strengths:

The data presented in the manuscript are of high quality and support the major conclusions.

Comments on revisions:

The authors have thoroughly addressed all the points raised during the first round of review. Their revisions effectively clarify key aspects of the manuscript, and the additional data and explanations have significantly improved the overall quality of the work. I believe the manuscript is now well-prepared for publication.

---

## [Author Response]

The following is the authors’ response to the original reviews

**Public Reviews:**

**Reviewer #1 (Public review):**
Summary:This manuscript by Kremer et al. characterizes the tissue-specific responses to changes in TFAM levels and mtDNA copy number in prematurely aging mice (polg mutator model). The authors find that overexpression of TFAM can have beneficial or detrimental effects depending on the tissue type. For instance, increased TFAM levels increase mtDNA copy number in the spleen and improve spleen homeostasis but do not elevate mtDNA copy number in the liver and impair mtDNA expression.Similarly, the consequences of reduced TFAM expression are tissue-specific. Reduced TFAM levels improve brown adipocyte tissue function while other tissues are unaffected. The authors conclude that these tissue-specific responses to altered TFAM levels demonstrate that there are tissue-specific endogenous compensatory mechanisms in response to the continuous mutagenesis produced in the prematurely aging mice model, including upregulation of TFAM expression, elevated mtDNA copy number, and altered mtDNA gene expression. Thus, the impact of genetically manipulating global TFAM expression is limited and there must be other determinants of mtDNA copy number under pathological conditions beyond TFAM.Strengths:Overall, this is an interesting study. It does a good job of demonstrating that given the multi-functional role of TFAM, the outcome of manipulating its activity is complex.Weaknesses:No major weaknesses were noted. We have minor suggestions for improving the clarity of the manuscript that are detailed in the "recommendations for the authors" section.

We thank the reviewer for the suggestions and addressed them as described in the "recommendations for the authors" section.

**Reviewer #2 (Public review):**
Summary:This study by Kremer et al. investigates the impact of modulation of expression of TFAM, a key protein involved in mitochondrial DNA (mtDNA) packaging and expression, in mtDNA mutator mice, which carry random mtDNA mutations. While previous research suggested that increasing TFAM could counteract the pathological effects of mtDNA mutations, this study reveals that the effects of TFAM modulation are tissue-specific. These findings highlight the complexity of mtDNA copy number regulation and gene expression, emphasizing that TFAM alone is not the sole determinant of mtDNA levels in contexts where oxidative phosphorylation is impaired. Other factors likely play a significant role, underscoring the need for nuanced approaches when targeting TFAM for therapeutic interventions.Strengths:The data presented in the manuscript is of high quality and supports major conclusions.Weaknesses:The statistical methods used are not clearly described, and some marked nonsignificant results appear visually significant, which raises concerns about data analysis.Data presentation requires improvement.

We thank the reviewer for the comments. We updated the text in the Materials and Methods section to state the statistical methods and improved the figures as described in detail in the "recommendations for the authors" section.

**Recommendations for the authors:**
(1) Please include testis data in Figure 2 given previous work by authors showing that elevated mtDNA copy number can improve testis function. It would be interesting to compare the changes in mtDNA copy number in testis to these other tissues.

We measured mtDNA copy number in testis using the CytB probe and added it as Supplementary figure 2 A.

(2) The clarity of Table 1 could be improved. It is difficult to know whether the changes in the TFAM to mtDNA ratio are driven by changes in TFAM levels or mtDNA copy number. A suggestion is to include the TFAM and mtDNA values in parenthesis next to each listed ratio.

We updated Table 1 and included the values of the normalized TFAM and mtDNA levels in parentheses.

(3) The authors should consider showing TFAM western blot data in Figure 1.

We thank the reviewer for the suggestion but would like to keep the TFAM western blot data with the other western blot data for the respective tissue.

(4) The graphs for qPCR data (e.g. Figure 2) show mRNA or mtDNA levels relative to the control, which is always set to 1. Why, then, does the control group display error bars?

For the normalization of the data to the WT group, we first calculate the average of the values from all the samples of the WT group. We then divide all values from the samples of all groups, including the WT group, by that average value. By doing so, we set the average value of the WT group to 1 and express all values from all samples of all groups, including the WT group, relative to this average value. Differences between the samples of the WT group are hence retained and allow for error calculations and the display of error bars.

(5) Page 3 second sentence to the last: overexpression of TFAM leads to...? Did the author mean mtDNA?

We updated the text to “Heterozygous knockout of *Tfam* in wild-type mice results in ~50% decrease of mtDNA levels, whereas moderate overexpression of *Tfam* leads to ~50% increase in mtDNA levels25,26”

(6) The sentence "In summary, mtDNA copy number regulation is more complex than previously assumed and the TFAM-to-mtDNA ratio seems to be finely tuned in a tissue-specific manner" - not clear who assumed (references?) and based on what data, please rephrase.

We updated the text and it now reads “In summary, mtDNA copy number regulation is more complex than suggested by previous studies23–27 and the TFAM-to-mtDNA ratio seems to be finely tuned in a tissue-specific manner.”

(7) The significant increase in complex II activity under TFAM overexpression (Figure 3) warrants additional discussion.

We updated the Results section and it now reads “We detected increased levels of the complex II subunit Succinate Dehydrogenase Complex Iron Sulfur Subunit B (SDHB). Complex II is exclusively nuclear encoded and a compensatory increase upon impaired mitochondrial gene expresson has been observed before32.

We proceeded to measure the enzyme activities of individual OXPHOS complexes in liver mitochondria (Fig. 3C). The complex I and complex IV activities were reduced to about 50% in *Polg*-/mut; *Tfam*+/+ mice in comparison with wild-type mice (Fig. 3C). However, we did not see any further alteration of the reduced enzyme activities induced by TFAM overexpression or reduced TFAM expression (Fig. 3C). Interestingly, we detected a significant increase in complex II and complex II + complex III activity upon TFAM overexpression, which can partially be explained by the increased complex II protein levels we oberseved in *Polg*-/mut; *Tfam*+/OE mice (Fig. 3, B and C).”

(8) The statistical methods used should be explicitly stated. Some results marked as non-significant appear visually significant, for example, mt-Cytb in Figure 2C, Supplementary Figure 2B.

We updated the text in the Materials and Methods section to state the statistical methods and it now reads “Statistical analysis and generation of graphs were performed with GraphPad Prism v9 software except for quantitative mass spectrometry data which was analyzed and plotted using R as described above. Statistical comparisons were performed using one-way analysis of variance (ANOVA), and post hoc analysis was conducted with Dunnett’s multiple comparisons test. Values of *P* < 0.05 were considered statistically significant.”

Minor points:(1) Replace numerical indications of significance with asterisks for consistency.

We replaced all numerical indications of significance with asterisks.

(2) Abbreviations SKM and BAT are not defined.

We removed the mentioning of SKM (skeletal muscle) as the data from this tissue was not included. The Introduction reads “In contrast, in brown adipose tissue (BAT), a decrease in TFAM levels normalized *Uncoupling protein 1* (*Ucp1)* expression.”

(3) Use uniform scales across bar graphs in Figure 2 to improve clarity.

We updated Figure 2 to have uniform scales.

(4) Remove or increase the transparency of data points in Figure 1A to make group averages more discernible.

We removed the data points in Figure 1A.

(5) Add a Y-axis title to Figure 1C.

We added the Y-axis title “Heart / body weight” to Figure 1C.

(6) Size of the font used in some figures (4?) is not appropriate.

We increased the font size for the figures.

(7) All figure legend titles need work. Insert "expression" after TFAM in the Figure 2 title, Change the title to "Modulation of TFAM expression..." in Figure 4.

The figure legends now read as follows:

“Figure 2: Modulation of TFAM expression affects mtDNA copy number in a tissue-specific manner.”

“Figure 4: Alteration of TFAM expression does not affect the heart phenotype of mtDNA mutator mice.”